# FactSpotter: Evaluating the Factual Faithfulness of Graph-to-Text Generation

**Kun Zhang**[1,2] and **Oana Balalau**[1,2] and **Ioana Manolescu**[1,2]
[1]Inria, [2]Institut Polytechnique de Paris

## Abstract

Graph-to-text (G2T) generation takes a graph as input and aims to generate a fluent and faithful textual representation of the information in the graph. The task has many applications, such as dialogue generation and question answering. In this work, we investigate to what extent the G2T generation problem is solved for previously studied datasets, and how proposed metrics perform when comparing generated texts. To help address their limitations, we propose a new metric that correctly identifies factual faithfulness, i.e., given a triple (subject, predicate, object), it decides if the triple is present in a generated text. We show that our metric FactSpotter achieves the highest correlation with human annotations on data correctness, data coverage, and relevance. In addition, FactSpotter can be used as a plug-in feature to improve the factual faithfulness of existing models. Finally, we investigate if existing G2T datasets are still challenging for state-of-the-art models. Our code is available online: https://github.com/guihuzhang/FactSpotter.

## 1 Introduction

Graph-to-text (G2T) generation is an important task in natural language generation, as it renders graphs, and in particular knowledge graphs, accessible to non-technical users in downstream applications such as question answering (Gu et al., 2021), (Romero and Razniewski, 2020), knowledge-grounded dialogue generation (Zhou et al., 2018), and document summarization (Fan et al., 2019). In recent years, there have been several datasets (Gardent et al., 2017; Nan et al., 2021) and methods proposed for G2T generation (Ke et al., 2021; Ribeiro et al., 2021), in addition to G2T competitions (Shimorina et al., 2018; Castro Ferreira et al., 2020). Evaluating text generation is a challenging task in itself (Celikyilmaz et al., 2020); moreover, in the context of G2T generation, we are concerned not only with the fluency of the gener-

ated text, but also with its faithfulness to the input graph. While recent models such as T5 and GPT models are very fluent, they have been criticized for their factual accuracy, a problem commonly referred to as hallucination (Ji et al., 2023; Liu et al., 2022). Hallucinations are a serious drawback in G2T, where the generated text should only contain facts mentioned in the input graph.

In this work, we focus on measuring and improving the factual accuracy of G2T generative models. More precisely, our contributions are as follows: *i*) We introduce a novel metric FactSpotter for detecting if G2T generations are faithful to the facts present in the input graph; *ii*) We show how FactSpotter can be used in the inference step of any G2T model to improve its generations; *iii*) We analyze the difficulty of existing G2T datasets and determine which are (resp., are no longer) challenging for state-of-the-art models. FactSpotter can be extended to other data-to-text tasks via methods for transforming a relational dataset into RDF, such as the R2RML language[1].

## 2 Related Work

### 2.1 Graph-to-text (G2T) generation

In (Ribeiro et al., 2021), the authors investigate the potential of pretrained language models (PLM) on the G2T task. They consider two Transformer-based models (Vaswani et al., 2017) with an encoder-decoder structure: T5 (Raffel et al., 2020), and Bart (Lewis et al., 2020). The models receive in input a linearized (serialized) version of the input graph, in which they add the tags $\langle H \rangle, \langle R \rangle$, and $\langle T \rangle$ before the head entity, the relation, and tail entity of a triple, respectively.

The potential of large language models is further investigated in (Keymanesh et al., 2022) on the DART dataset (Nan et al., 2021), a data-to-text dataset constructed from tables, available in

---

[1]http://www.w3.org/TR/r2rml/

a triple-to-sentence format. The dataset is constructed from a combination of manual and automatic techniques. The authors empirically evaluate the GPT2 model (Radford et al., 2019) and the T5 model on the dataset by varying the amount of supervision a model receives. They also investigate the potential of adding predicate descriptors in the prompt and re-ranking generations. In a small-scale human evaluation, they find that their best model, T5 large, outperforms the reference text regarding hallucinations and fluency, underlining that *existing datasets suffer from poor human annotations*, which we also observe and discuss in Section 7.

The authors of (Ke et al., 2021) propose modifying the Transformer model by adding extra attention and pooling layers to improve G2T generation. In addition, the model's pretraining has three steps given the input graph and the expected output text: 1) reconstructing the text sequence given the complete subgraph, 2) predicting the masked entities and relations in the corrupted subgraph, given the complete text, and, 3) aligning the embedding vectors of the knowledge graph and the text.

Most state-of-the-art G2T models are based on Transformers, and they can generally generate fluent texts related to given graphs. Although various baselines designed neural networks to encode both global and local information (Ribeiro et al., 2020), they cannot guarantee that generated texts are factual faithful to the given graphs. It's also not clear whether current G2T datasets are still challenging.

## 2.2 Evaluation metrics for text generation

Alongside the significant improvements that models for G2T generation underwent and, in general, the improvement of language models, new metrics to assess the generations' quality have been proposed (Celikyilmaz et al., 2020; Sai et al., 2022). G2T generation belongs to the broader field of natural language generation, including tasks such as machine translation, automatic summarization, question answering, and more. Each task has specific requirements, which might entail using some metrics over others. For example, in machine translation, the translation should match the ground-truth text as closely as possible, while in chatting or summarization, adding or removing some information is acceptable or even desirable.

Evaluation metrics are split into three categories: *human-centric* metrics, *untrained automatic* metrics, and *machine-learned* metrics. Human evaluation is the most important of these metrics. It consists of asking users to evaluate the quality of a text in a specific category, such as fluency or correctness. Unfortunately, human evaluation sometimes suffers from a low inter-annotator agreement (Celikyilmaz et al., 2020), (Belz and Reiter, 2006) as different people might have different notions of what makes a text fluent or correct, or the instructions they receive for annotation might lack sufficient clarity. Also, human annotation is time and money-consuming; it can represent a bottleneck in the iterative process of improving a model. Hence the need for automatic metrics such as BLEU (Papineni et al., 2002), ROUGE (Lin, 2004), METEOR (Banerjee and Lavie, 2005), BERTScore (Zhang et al., 2019), BLEURT (Sellam et al., 2020) BARTScore (Yuan et al., 2021), InfoLM (Colombo et al., 2022), among others. Overlap measures based on $n$-*grams*, such as BLEU, ROUGE, and METEOR, have been widely used in the literature, while recently proposed metrics based on *word embeddings*, such as BERTScore, BLEURT, BARTScore are gaining traction. The embedding-based measures have been shown to correlate better with human evaluation than the n-grams metrics. In addition, some metrics, such as BLEURT, have been trained on human annotations. As these automatic metrics are based on distances or similarities to ground-truth texts, they rely on the quality of annotated sentences.

Apart from the works mentioned above, a few prior studies assessed factual faithfulness in graph-to-text generation. In (Faille et al., 2021) the authors introduce a metric for verifying whether entities in input graphs are represented in the generated texts. However, this work does not evaluate the quality of the predicates in the generated texts, which is a much more difficult task. In (Rebuffel et al., 2021), a question generation (QG) and question answering (QA) framework is employed to evaluate the quality of the generated text by determining if each question posed by the QG module can be addressed by the QA module. We believe our contribution can further advance the state-of-the-art as: *i*) FactSpotter requires significantly less computational resources; *ii*) FactSpotter is self supervised, thus it does not requires additional data to the G2T model; item FactSpotter can be pluged into a G2T to improve its generation.

## 3   Problem Statement

A knowledge graph (KG) consists of facts (or data triples) of the form ⟨subject, relation, object/literal⟩ and/or an ontology describing the properties that hold between classes and/or relations.

**Graph-to-text.**   A G2T tool takes in input a graph and outputs a textual representation of the graph. G2T inputs are often subgraphs of larger real-world graphs, such as knowledge graphs. The textual representation should be fluent and should contain all the facts present in the input graph. For example, given a subgraph consisting of the single DBpedia (Auer et al., 2007) fact ⟨The Myth of Sisyphus, author, Albert Camus⟩, we would like to generate the sentence *The Myth of Sisyphus was written by Albert Camus*. This work primarily focuses on creating textual representations of KG subgraphs.

**Factual faithfulness.**   The following human criteria have been proposed to evaluate the factual quality of a generated text (Castro Ferreira et al., 2020): data correctness, data coverage, and relevance. Given a graph $G$ and a machine-generated text $T$, the generated text is characterized by:

1. *Data coverage* or *recall*: are all the descriptions presented in the graph $G$ included in the text $T$? *i*) *Covered predicates:* does $T$ contain all the predicates from $G$? *ii*) *Covered entities:* does $T$ contain all the entities from $G$?

2. *Relevance* or *precision*: *i*) *Relevant predicates:* does $T$ only contain relevant predicates? *ii*) *Relevant entities:* does $T$ only contain relevant entities?

3. Correctness: are predicates correctly mentioned and adequately introduced in the data?

**Research questions.** We focus on the following three research questions:

**RQ1** What metric would better correlate with the factual faithfulness of generated text?

**RQ2** Can we improve factual faithfulness of G2T?

**RQ3** Is the G2T task solved on existing datasets?

## 4   FactSpotter: An explainable metric for factual faithfulness

In this section, we introduce a new metric for factual faithfulness. A good metric should be **interpretable**, that is: given in input a fact $f$ of the form ⟨subject, predicate, object⟩ and a text $T$, it assigns a score between 0 and 1, where a score close to 0 signifies that the text that does not correctly represent the fact, and a score close to 1 rewards a factual faithful textual representation of $f$ in $T$. Such a metric can be used to compare different generation systems and, in addition, assess a single system on a given dataset, to determine how close the system is to representing the input graphs correctly.

The **intuition of our score** is the following. We train a model to perform a task simpler than G2T generation: only *detecting whether facts exist in a sentence* and *whether they are well expressed*. This simpler model can then be used as a **plug-in feature** by any existing G2T model, to aid it perform the more complex task of language generation.

Our metric, FactSpotter, is trained as a binary classifier. Given in input a fact ⟨subject, predicate, object⟩ and a sentence, it should predict 1 if the fact is well expressed in the sentence, or 0 otherwise. Thus, FactSpotter is inherently interpretable. We leverage as classification models recent large language models, capable of detecting semantic closeness, even if different words, e.g., synonyms, are used. This approach is similar to the one taken to compute metrics such as BertScore and BartScore. Given an input G2T dataset $D$, with a *training* (train) set, a *development* (dev) set, and a *test* set, we create the training set as follows:

**Positive samples.**   Given an instance of the training set of the form (graph $G$, ground truth text $T$), for each fact (triple) in $G$, we create a positive sample of the form (fact $f$, ground truth text $T$).

**Negative samples.**   Given an instance of the training set of the form (graph $G$, ground truth text $T$), for each fact $f \in G$, we create negative samples as follows: *i*) Type I: we perturb the fact $f$: we change its predicate, or an entity (subject or object), or both, while the ground truth text $T$ remains unchanged. *ii*) Type II: we perturb the ground truth text $T$: we drop one or both entities related to $f$ from the text, or drop the $n$-grams most similar to the predicate of $f$, or we apply simultaneously several modifications, keeping the fact unchanged.

For example, given the fact ⟨The Myth of Sisyphus, author, Albert Camus⟩ and its associated text "The Myth of Sisyphus was written by Albert Camus", a Type I negative sample alters the fact (⟨The Myth of Sisyphus, author, *Simone de Beauvoir*⟩, "The Myth of Sisyphus was written by Albert Ca-

mus"), while for Type II yields the sample ( ⟨The Myth of Sisyphus, author, Albert Camus⟩, "The Myth of Sisyphus was written"). We associate probabilities to each perturbation, and control the generation such that for each positive sample, we only generate one negative sample. To allow our classifier to learn from different negative samples and avoid over fitting (Chen et al., 2020), for each training epoch, we use a newly generated set of negative samples. The development set is built in the same way. Through evaluation on a fixed test set (Appendix A.4.1) we find that the model which best detects factual faithfulness is the one with the highest probability of perturbing the fact (90%) and 10% the probability of perturbing the ground truth.

Above, we have described FactSpotter as a *(trained) classifier*. To use it as a *score (metric)*, we take the output of the model after the softmax layer. The final score of a generated text $T$ given an input graph $G$ is the average over the scores for each pair (fact $f \in G$, generated text $T$).

**Parameters and performance of FactSpotter.** As we aim to add our metric, FactSpotter, in the inference step of graph-to-text generation, we prefer small language models. Hence, we select the small Electra model (Clark et al., 2020). We have experimented with other small models, such as DistilBERT and DistilRoBERTa, but we did not observe an improvement. We train our classifier for 16 epochs, with a learning rate of $5 \cdot 10^{-5}$, and the AdamW optimizer. We describe in the Appendix A.4.1 how we chose the percentages of negative samples for FactSpotter.

The performance of FactSpotter on the test splits across multiple datasets is detailed in Table 1, with accuracy and F1 score. We also report numbers of true positives/negatives (TP/TN), and false positives/negatives (FP/FN) in the table.

| Dataset | Acc. | F1 | TP/TN | FP/FN |
|---------|------|------|-------------|--------|
| GrailQA | 96.85 | 96.85 | 2272/2643 | 64/96 |
| SimpleQ. | 95.41 | 95.41 | 10260/10438 | 419/575 |
| DART | 97.62 | 97.62 | 26613/26352 | 607/684 |
| WebNLG17 | 99.10 | 99.09 | 8594/8575 | 93/63 |
| WebNLG20 | 95.21 | 95.21 | 9637/9853 | 317/663 |

Table 1: Performance of FactSpotter on test splits.

## 5 Evaluating graph-to-text generation

We investigate our first research question (**RQ1**): which metrics would better correlate with the fac-

tual faithfulness of the generated text? For this question, we compare FactSpotter with: BLEU (Papineni et al., 2002), METEOR (Banerjee and Lavie, 2005), BERTScore (Zhang et al., 2019), BLEURT (Sellam et al., 2020), BARTScore (Yuan et al., 2021). The only metric that is not normalized is BARTScore. This metric has a variant specifically adapted for factual faithfulness, BARTScore-faithful. Further details of these metrics are provided in Appendix A.1.

We calculate the *system-level correlation* between automatic metrics and human annotations. Given $S$ systems under evaluation, for a certain dimension, e.g., fluency, we compute the correlation between the system-level *automatic metric scores* $[M(S_1), \ldots M(S_S)]$ and the corresponding *system-level human scores* $[H(S_1), \ldots H(S_S)]$, where $M(S_i)$ is the score of the automatic metric on the texts from system $S_i$, and $H(S_i)$ is the score of the human annotation on the same result. Similarly to (Colombo et al., 2022), we compute three correlation metrics: Pearson correlation ($r$), Spearman correlation ($\rho$), and Kendall's Tau ($\tau$). To test if a metric $M_1$ has a higher correlation with human annotations than $M_2$, we use the bootstrapping technique proposed in (Wilcox, 2016), which we describe in the Appendix A.2. We also report the sentence-level correlation in Appendix A.5. In addition to automatic measures, we report the correlation between one annotator and the average score of the remaining annotators, which should be a upper bound on the correlation we can obtain using automatic measures.

**WebNLG 2017.** In the WebNLG 2017 challenge (Shimorina et al., 2018), the organizers annotated 9 submissions on *semantic adequacy* (the text correctly represents the meaning in the data), *text structure* (as above, referred in the original paper as grammar) and *fluency* (as above). This annotation has carried over 223 samples.

**WebNLG 2020.** After the WebNLG 2020 Challenge (Castro Ferreira et al., 2020), the organizers annotated the 16 participating systems on *data correctness* (the predicates found in the data are correctly mentioned together with their subject and object), *data coverage* (the text includes descriptions of all predicates presented in the data), and *relevance* (the text describes only those predicates with related subjects and objects which are in the data), in addition to *text structure* (the text is grammati-

| Metric | Correct. | | | D. Cover. | | | Relev. | | | Fluency | | | T. Struct. | | |
|---|---|---|---|---|---|---|---|---|---|---|---|---|---|---|---|
| | $r$ | $\rho$ | $\tau$ | $r$ | $\rho$ | $\tau$ | $r$ | $\rho$ | $\tau$ | $r$ | $\rho$ | $\tau$ | $r$ | $\rho$ | $\tau$ |
| Correct. | 1.0 | 1.0 | 1.0 | 0.96 | 0.81 | 0.66 | 0.97 | 0.81 | 0.66 | 0.80 | 0.77 | 0.60 | 0.79 | 0.76 | 0.59 |
| D. Cover. | | | | 1.0 | 1.0 | 1.0 | 0.93 | 0.80 | 0.64 | 0.71 | 0.56 | 0.43 | 0.69 | 0.55 | 0.42 |
| Relev. | | | | | | | 1.0 | 1.0 | 1.0 | 0.76 | 0.63 | 0.48 | 0.76 | 0.62 | 0.47 |
| Fluency | | | | | | | | | | 1.0 | 1.0 | 1.0 | 0.98 | 0.97 | 0.91 |
| T. Struct. | | | | | | | | | | | | | 1.0 | 1.0 | 1.0 |
| Human | 0.96 | 0.80 | 0.65 | 0.93 | 0.83 | 0.68 | 0.96 | 0.74 | 0.59 | 0.95 | 0.93 | 0.80 | 0.93 | 0.91 | 0.77 |
| BLEU | 0.59 | 0.64 | 0.48 | 0.53 | 0.53 | 0.40 | 0.56 | 0.60 | 0.45 | 0.87 | 0.84 | 0.68 | 0.86 | 0.84 | 0.68 |
| METEOR | 0.72 | 0.75 | 0.60 | 0.65 | 058 | 0.44 | 0.70 | 0.64 | 0.50 | 0.88 | 0.89 | 0.74 | 0.86 | 0.88 | 0.72 |
| BERTF1 | 0.83 | 0.77 | 0.60 | 0.74 | 0.58 | 0.43 | 0.81 | 0.65 | 0.50 | **0.90** | **0.93** | **0.80** | 0.88 | **0.92** | **0.78** |
| BLEURT | 0.93 | 0.82 | **0.67** | 0.86 | 0.65 | 0.50 | 0.91 | 0.69 | 0.55 | **0.90** | 0.92 | 0.78 | **0.89** | 0.91 | 0.76 |
| BARTS | 0.90 | **0.83** | **0.67** | 0.86 | 0.71 | 0.53 | 0.88 | 0.71 | 0.56 | 0.77 | 0.81 | 0.63 | 0.75 | 0.80 | 0.62 |
| BARTS-F | 0.67 | 0.54 | 0.41 | 0.68 | 0.61 | 0.46 | 0.68 | 0.59 | 0.45 | 0.51 | - | - | 0.52 | - | - |
| FactS | **0.94** | 0.80 | 0.64 | **0.91** | **0.87** | **0.71** | **0.96**\* | **0.79** | **0.64** | 0.74 | 0.59 | 0.45 | 0.72 | 0.59 | 0.45 |

Table 2: Correlation at the system level with human judgment on correctness, data coverage, relevance, fluency and text structure for the 2020 WebNLG task. For tables here and below, BERTF1 stands for BERTScore-F1, BARTS for BARTScore, BARTS-F for BARTScore-faithful, FactS for FactSpotter. We **highlight** the best result and we mark it with an asterisk when it is statistically significantly larger than any other metric (excluding Human to Human correlation). FactSpotter performs the best on correctness, data coverage, and relevance. We put a value for correlation if the pvalue $p < 0.05$.

| Metric | Sem. Adeq. | | | T. Struct. | | | Fluency | | |
|---|---|---|---|---|---|---|---|---|---|
| | $r$ | $\rho$ | $\tau$ | $r$ | $\rho$ | $\tau$ | $r$ | $\rho$ | $\tau$ |
| Sem. Adeq. | 1.0 | 1.0 | 1.0 | 0.73 | 0.65 | 0.52 | 0.71 | 0.66 | 0.52 |
| T. Struct. | | | | 1.0 | 1.0 | 1.0 | 0.98 | 0.95 | 0.88 |
| Fluency | | | | | | | 1.0 | 1.0 | 1.0 |
| Human | 0.99 | 0.98 | 0.94 | 0.98 | 0.92 | 0.84 | 0.98 | 0.89 | 0.79 |
| BLEU | 0.76 | 0.71 | 0.56 | **0.86** | **0.71** | **0.57** | **0.83** | **0.72** | **0.57** |
| METEOR | 0.86 | 0.83 | 0.67 | 0.85 | 0.70 | **0.57** | 0.80 | 0.70 | 0.56 |
| BERTF1 | 0.70 | 0.78 | 0.63 | 0.71 | 0.70 | **0.57** | 0.69 | 0.70 | **0.57** |
| BLEURT | 0.90 | 0.88 | 0.72 | 0.84 | 0.69 | 0.56 | 0.79 | 0.68 | 0.56 |
| BARTS | 0.90 | 0.87 | 0.78 | 0.71 | - | - | 0.68 | - | - |
| FactS | **0.97**\* | **0.93** | **0.85** | 0.67 | - | - | - | - | - |

Table 3: Correlation at the system level with human judgment on semantic adequacy, grammar, and fluency, for the 2017 WebNLG dataset.

cal, well structured, and written in good English) and *fluency* (the text progresses naturally, forms a coherent whole and is easy to understand). 178 generations of each system were annotated.

**Results.** On WebNLG 2020, Table 2 shows that FactSpotter has the best performance on factual faithfulness, significantly improving *relevance*. BLEU, METEOR, BERTScore and BLEURT reach similar *fluency* and *text stucture* scores. For the results on WebNLG 2017 in Table 3, FactSpotter has the highest performance on *semantic adequacy*, which is the only dimension related to factual faithfulness. For *text structure* and *fluency*, BLEURT obtains the best results, although the results are not statistically significant. Overall, previous metrics are better on *text structure* and *fluency*, which are generally considered as no longer a bottleneck for large language models. FactSpotter is the best suited metric on factual faithfulness.

# 6 Improving the factual faithfulness of graph-to-text generation

In this section, we investigate the answer to our third research question: can we improve graph-to-text generation on factual faithfulness (**RQ2**)? For this, we first explain how to improve the inference step of any G2T model using FactSpotter, and then we present the results of this technique on the state-of-the-art models for G2T generation.

## 6.1 Improving models' factual faithfulness

Let $\mathcal{M}$ be a neural network G2T (seq2seq) model that, given an input sequence $x = x_1, x_2, \ldots x_M$, produces an output sequence $y = y_1, y_2, \ldots, y_N$, where $x_i \in V_x$, $y_i \in V_y$, and $V_x$, $V_y$ are the vocabularies from where we select items in the input, respectively output sequence. In the inference step, the model generates the sequence $y$ that maximizes:

$$P(y|x) = \prod_{i=1}^{N} P(y_i|y_{<i}, x)$$

In practice, for computational efficiency, the $\log$ of the probabilities are typically utilized in beam search. We use the following method to improve factual faithfulness in G2T inference *without retraining the model*.

Given: *i*) a graph-to-text generation model $\mathcal{M}$, *ii*) our factual faithfulness classifier, i.e., FactSpotter, *iii*) a subgraph $G$ composed of $F$ facts, we encourage factual generations by modifying the

prediction step as follows:

$$\log(P^f(y_i|y_{<i}, x)) =$$
$$\lambda \sum_{j=1}^{F} (1 - P_{fact_j}(y_{<i-1})) \log P_{fact_j}(y_{<i})$$
$$+ \log(P(y_i|y_{<i}, x)) \quad (1)$$

where: *i*) $P^f(y_i|y_{<i}, x)$ is the probability of generating token $y_i$ given the factual classifier; *ii*) $P_{fact_j}(y_{<i-1})$ is the probability of correctly representing the fact $j$ in the previously generated tokens $y_0, ..., y_{i-1}$, computed by FactSpotter. *iii*) $P(y_i|y_{<i}, x)$ is the probability for generating the next token based on previous $i - 1$ tokens.

In our equation, a fact $j$ is encouraged only if we have not observed in the text generated until step $i$, according to $P_{fact_j}(y_{<i-1})$. Then adding token $y_i$ would increase the probability of the text including the fact $j$, $P_{fact_j}(y_{<i})$. When $P_{fact_j}(y_{<i-1})$ is small, the equation encourages the selection of words belonging to the fact $j$. As $P$ is large and $1 - P_{fact_j}(y_{<i-1})$ tends to 0, then we can consider that the fact $j$ already appears in the text, and words that satisfy fact $j$ will no longer be encouraged. The weight $\lambda$ controls the influence of the FactSpotter on the prediction of the following words. A high $\lambda$ might yield a factual text, but not necessarily a fluent one. We generate tokens till we have generated text $S = y_0, ..., y_k$ for which $\forall j \in F, P_{fact_j}(S) > 0.5$, i.e., the probability of each fact $j$ in $G$ is verbalized in the text $S$ is over 0.5, the standard positive threshold.

## 6.2 Models

We consider for our evaluation state-of-the-art models for G2T genetation, PLM (Ribeiro et al., 2021) and JointGT (Ke et al., 2021). The former investigates how a standard seq2seq model can perform on G2T, given a carefully constructed representation of a graph. The latter proposes a more complex neural network, with built-in attention layers specialized on graph structured inputs. Both are initialized with the pretrained weights of a language model. Similar to the authors, we consider in this work T5 (Raffel et al., 2020) for both G2T models. For simplicity we refer to the first model as **T5**, and to the second model as **JointGT**. We refer to the models modified as explained in Section 6.1 as **FactT5** and **FactJointGT**. For T5 and FactT5, we initialize the weights with T5 small, and T5 base. For JointGT and FactJointGT we initialize

the weights with a pretrained T5 base model offered by the authors of JointGT. For the fine-tuning step (each model is fine-tuned on the training split of the datasets), we train the small models with a learning rate of $10^{-4}$ and a batch size of 32, while the base models are trained with a learning rate of $5 \cdot 10^{-5}$ and a batch of 16. We use a beam size of 5 and the AdamW optimizer for both sizes of models. For Equation 1, we fix the weight $\lambda = 0.15$ (parameter tuning in Appendix A.4.2).

## 6.3 Datasets

To evaluate G2T performance, we need *(graph, text)* pairs datasets. The graphs can be directly extracted from an underlying knowledge graph or adapted to have richer semantics, such as query graphs (Yih et al., 2015). The text associated to the graph should be a sentence or a paragraph verbalizing all the information contained in the subgraph.

Several datasets are proposed in the literature for the G2T task, such as WebNLG (Gardent et al., 2017). Many question-answering (QA, in short) datasets are also in the form (graph, text). Since question answering datasets can be very large and cover many KG predicates (Gu et al., 2021), we also leverage such datasets. To ensure that FactSpotter has never encountered the test data, it is trained exclusively on the training set and evaluated it on the validation split.

**SimpleQuestions** (Bordes et al., 2015) is a QA dataset built on Freebase (Bollacker et al., 2008). The dataset contains 108K (triple, question) pairs, where the question corresponds to the subject and predicate from the triple, and the answer is the object of the triple. For example, given the triple *(Gulliver's Travels, book/written_work/author, Dean Swift)*, the question is *Who wrote Gulliver's Travels?*, with the answer Dean Swift. The dataset covers 76 domains, 741 classes, 89K entities and 2K relations. A Freebase domain is a general area of knowledge such as business, politics, economics, etc. We created our own split for this dataset, where the test set is zero shot: we have not seen the predicates during training. FactSpotter can be trained to correctly classify if a question refers to a triple, even if the object or subject is missing from the question, as we replace the entity with its type.

**GrailQA** (Gu et al., 2021) is also a QA dataset that uses Freebase, created using human annotation. The original dataset contains $64K$ (triple, question) pairs, however, the test set is not released

as the authors have created a QA challenge [2], hence we use the development set as a test set. The remaining data (training and development) consists of 51K pairs. The authors propose three levels of generalization and split the development and test as follows. 50% of the pairs from held-out domains (Freebase assigns to each entity and predicate a domain, such as music, sports, etc.) are not covered in training: this is the zero-shot split. 25% of the pairs correspond to graphs where the combination of ontology items (classes and predicates) were not covered in training: this is the compositional split. Finally, the remaining 25% are randomly sampled from the same distribution as the training dataset: the i.i.d. split. The i.i.d. and compositional subsets contain only ontology items (classes and predicates) covered in training. For the zero-shot subset, five domains are held out for validation.

**WebNLG** (Gardent et al., 2017; Castro Ferreira et al., 2020) is a text generation dataset on DBPedia, created via human annotation. The dataset consists of (graph, paragraph) pairs, where the graph is a set of DBPedia facts, and the paragraph consists of one or several sentences that describe the graph. We use the 2017 (2.1) and 2020 (3.0) versions of the dataset [3]. The 2017 version of the dataset contains 42K graph-text pairs, and it has two splits, the standard version and the constrained version. In the constrained version, the test set does not contain that a triple occurring in train/dev. In this work we considered only the constrained split, as it is more challenging. The WebNLG 2020 dataset has 40K pairs, which comprises 10 categories that were previously seen and utilized in WebNLG 2017, as well as 5 categories that were unseen in WebNLG 2017 are now incorporated into the seen data of the WebNLG 2020 dataset. It also introduces a new category of data: company.

**DART** (Nan et al., 2021) is a data-to-text dataset based on Wikipedia tables. Since the tables are represented as (subject, predicate, object) triples, it also suits our evaluation. Besides creating table-to-text annotations, the authors also use existing datasets: the QA dataset WikiSQL (Zhong et al., 2017), the cleaned E2E (Dušek et al., 2019) (entity-to-entity relations in the restaurant domain), and the original release of the WebNLG dataset for the 2017 challenge (Shimorina et al., 2018). The au-

thors align the predicates such that predicates with the same meaning have the same representation. The dataset has 82K instances. We excluded the WikiSQL split as it has been generated automatically and after we performed a manual verification, we observed many low quality ground truth texts.

## 6.4 Evaluation

In this section, we consider (**RQ2**): Can we improve G2T generation on factual faithfulness?

Table 4 shows the results on the SimpleQuestions dataset. We generally have a high FactSpotter score, indicating that models are already good at relaying factual information. We can improve the factual faithfulness with F-T5 and FGT without significant compromise on other metrics, implying maintained fluency of texts.

| Model | BLEU | METEOR | BERTF1 | BLEURT | BARTS | FactS |
|-------|------|--------|--------|--------|-------|-------|
| T5S | 37.97 | 36.50 | 93.43 | 67.85 | -2.45 | 96.80 |
| F-T5S | 37.85 | 36.06 | 93.42 | 67.86 | -2.45 | **98.17** |
| T5B | 38.73 | 36.43 | 93.61 | 68.48 | -2.43 | 95.09 |
| F-T5B | 38.73 | 36.42 | 93.56 | 68.42 | -2.43 | 97.14 |
| JGT-T5 | **39.35** | **36.82** | **93.65** | **68.50** | **-2.42** | 95.40 |
| FGT-T5 | 39.24 | 36.78 | 93.64 | 68.48 | **-2.42** | 97.25 |

Table 4: Results on G2T on SimpleQuestions. Here and below, T5S stands for T5 small, T5B for T5 base, F-T5S for FactT5 small, F-T5B for FactT5 base, JGT-T5 for JointGT-T5, and FGT-T5 for FactJointGT-T5.

| Model | BLEU | METEOR | BERTF1 | BLEURT | BARTS | FactS |
|-------|------|--------|--------|--------|-------|-------|
| T5S | 66.24 | 47.80 | 96.72 | 73.16 | -1.41 | 98.67 |
| F-T5S | 66.27 | 47.89 | 96.73 | 73.21 | -1.41 | 99.25 |
| T5B | 67.04 | **48.35** | **96.81** | 73.22 | -1.40 | 99.44 |
| F-T5B | 67.04 | 48.22 | 96.80 | 73.26 | -1.40 | **99.71** |
| JGT-T5 | **67.08** | 48.34 | 96.76 | **73.44** | **-1.39** | 99.09 |
| FGT-T5 | 66.89 | 48.19 | 96.84 | 73.42 | **-1.39** | 99.67 |

Table 5: Results on G2T on WebNLG 2017 Const.

| Model | BLEU | METEOR | BERTF1 | BLEURT | BARTS | FactS |
|-------|------|--------|--------|--------|-------|-------|
| T5S | 52.30 | 40.82 | 93.43 | -1.75 | 65.80 | 90.75 |
| F-T5S | 52.44 | 41.02 | 93.45 | -1.74 | 65.92 | 93.45 |
| T5B | 54.29 | 41.66 | **93.65** | **-1.69** | 66.43 | 93.60 |
| F-T5B | **54.72** | **41.70** | 93.61 | **-1.69** | **66.46** | **95.14** |
| JGT | 54.23 | 41.49 | 93.47 | -1.72 | 66.23 | 91.26 |
| FGT-T5 | 54.45 | 41.52 | 93.49 | -1.72 | 66.31 | 93.16 |

Table 6: Results on G2T on the WebNLG 2020 dataset.

| Model | BLEU | METEOR | BERTF1 | BLEURT | BARTS | FactS |
|-------|------|--------|--------|--------|-------|-------|
| T5S | 46.22 | 39.96 | 94.69 | 66.62 | -2.03 | 95.47 |
| F-T5S | 46.31 | 40.07 | 94.74 | 66.66 | -2.02 | 97.29 |
| T5B | **48.47** | **40.74** | 95.04 | **67.49** | -1.97 | 96.65 |
| F-T5B | 48.37 | 40.72 | **95.05** | 67.43 | -1.97 | **97.60** |
| JGT-T5 | 47.51 | 40.43 | 94.92 | 67.33 | -2.01 | 95.86 |
| FGT-T5 | 47.39 | 40.32 | 94.92 | 67.26 | -2.00 | 97.25 |

Table 7: Results on G2T on the DART dataset.

[2]https://dki-lab.github.io/GrailQA/
[3]https://gitlab.com/shimorina/webnlg-dataset

| Model/Split | BLEU | METEOR | BERTF1 | BLEURT | BARTS | FactS |
|---|---|---|---|---|---|---|
| **IID** | | | | | | |
| T5S | 44.51 | 41.80 | 93.23 | 69.53 | -2.37 | 97.98 |
| F-T5S | 44.64 | 41.88 | 93.25 | 69.63 | -2.36 | 98.47 |
| T5B | 45.95 | 42.71 | 93.50 | 70.66 | **-2.29** | 99.43 |
| F-T5B | **46.10** | **42.67** | **93.52** | **70.73** | **-2.29** | **99.50** |
| JGT-T5 | 43.68 | 41.65 | 93.21 | 69.41 | -2.37 | 98.62 |
| FGT-T5 | 43.61 | 41.65 | 93.19 | 69.41 | -2.37 | 99.12 |
| **Zero** | | | | | | |
| T5S | 30.30 | 36.91 | 91.74 | 62.87 | -2.74 | 93.27 |
| F-T5S | 30.30 | 36.90 | 91.75 | 63.01 | -2.73 | 94.60 |
| T5B | 32.20 | 37.35 | 91.92 | 63.84 | -2.72 | 94.77 |
| F-T5B | 32.39 | 37.39 | 91.92 | 63.94 | -2.71 | **95.61** |
| JGT-T5 | **32.94** | **37.69** | **92.02** | **64.18** | -2.68 | 94.15 |
| FGT-T5 | 32.46 | 37.55 | 91.93 | 64.00 | -2.68 | 94.95 |
| **Comp.** | | | | | | |
| T5S | 30.38 | 35.32 | 92.09 | 63.99 | -2.74 | 94.94 |
| F-T5S | 30.14 | 35.21 | 92.08 | 63.72 | -2.75 | 96.58 |
| T5B | **31.75** | 35.64 | 92.24 | 63.99 | -2.72 | 94.84 |
| F-T5B | 31.66 | **35.72** | 92.24 | 64.10 | -2.71 | 96.53 |
| JGT-T5 | 31.46 | 36.08 | 92.39 | 64.92 | -2.67 | 95.26 |
| FGT-T5 | 31.25 | 36.21 | **92.43** | **65.12** | **-2.65** | **97.10** |

Table 8: Results on G2T on the GrailQA dataset.

Table 5 has the highest FactSpotter score from all datasets, which means that we observe the most factual generations on WebNLG 2017, with F-T5 and FGT having slightly higher scores.

In Table 6, the FactSpotter scores are lower for the WebNLG 2020 test split, although we achieve scores comparable to WebNLG 2017 on its validation split. This discrepancy may be attributed to the difference in distribution between the test and training splits of WebNLG 2020. F-T5B can achieve higher FactSpotter score than T5B without compromising fluency. We observe the same trends for the DART dataset in Table 7.

In Table 8, all the metrics are higher for the IID split of GrailQA, and in particular FactSpotter can reach 99.5%, hence the models learn to reproduce triples seen in training. For Zero-shot and Compositional splits, larger models are better, and our factual inference improves the score of FactSpotter. We illustrate some improved samples of Zero-shot and Compositional splits in Appendix A.3, and in Appendix A.6, we investigate the impact of varying numbers of triples in the input subgraphs on the quality of the generated text.

To validate that indeed generations improve using FactSpotter in inference, we select the best FGT-T5 model and we analyse the top 20 phrases where the FactSpotter score improved the most compared to the JGT-T5 generations.

For the SimpleQuestion datasets, we have 15 generations that are more factual, and 5 generations less factual. For the Zero-shot split of GrailQA, 14 generations are more factual. For its Composi-

tional split, we have 13 improved generations, but also 5 that are less factual. For the IID split, 4 generations are improved, others are all rephrased texts. Only 7 samples of IID improve over 0.01 for FactSpotter, so this split is not challenging. For the DART dataset, 6 texts are more factual. DART dataset has samples that ground-truth sentences do not match with graphs, so FactSpotter trained on DART has false positives. For WebNLG 2017 dataset, 11 generations are more factual, others are rephrased texts. WebNLG 2017 is only has 16 generations improve over 0.1 for FactSpotter, whose baseline is very high. For WebNLG 2020, 12 generations are more factual, and 3 are rephrased texts. 5 generations in WebNLG 2020 have higher FactSpotter than baseline generations, but they're still not factual enough.

For the cases where a generation becomes less factual, this is a consequence of the accuracy of FactSpotter, which we present in Section 4. Given that our metric does not correlate strongly with fluency, we perform a second analysis on generations to observe if there is a decrease in fluency. To answer this question, we study the top 20 sentences for which the BLEURT decreased the most in comparison with the original generated question. We do not observe an obvious decrease in fluency on any dataset, the decrease in BLEURT score is due to several other factors: BLEURT has difficulties identifying rephrased sentences, in a few cases the factual faithfulness decreased, and in the remaining cases the generations are more faithful to the input graph than the ground truth sentences, however BLEURT cannot identify it. Hence, we can conclude that adding FactSpotter as a plugin in generation can improve G2T generations on factual faithfulness and does not affect the fluency.

## 7 Remaining challenges in G2T task

Finally, we consider **(RQ3)**: is G2T task solved on existing datasets? We have observed high FactSpotter scores in Section 6 on the performances of models. We use FactSpotter to do a more detailed analysis: we investigate what is the percentage of generations in each dataset which had at least a fact considered missing by FactSpotter. A fact is considered as missing if the score of the pair (fact, generated sentence) is lower than 0.5. We obtain the following statistics: 1.94% of texts miss a fact in SimpleQuestions; 7.27% of texts miss at least a fact in DART; 5.79% of WebNLG 2017 texts

miss at least one fact, and 12.64% for WebNLG 2020; For GrailQA we have 5.8% for the zero shot split, 4.36% for the compositional split and 1.13 for the IID split. According to the observations, WebNLG 2020 is the most challenging dataset, followed by DART, the zero shot split of GrailQA, and the WebNLG 2017 dataset. In Appendix A.7, taking GrailQA and WebNLG 2017 as examples, we analysed the difficulty of G2T on datasets from different knowledge graphs, by looking into how often predicates and entity names are rephrased or expressed exactly as in the input graph.

We perform a second evaluation, this time by manually analyzing the output of the models. We consider the worst 20 sentences according to BLEURT and `FactSpotter`, hence 40 examples per dataset or split. On SimpleQuestions, the generations are fluent, however 22/40 have an incorrect predicate. For GrailQA, in the IID split the predicates are correctly generated, but the models still have difficulties in generating some entity names (16/40). For the zero-shot split, generations suffer from wrong entities and predicates (22/40). The compositional split has several samples with wrong ground truth (6/40 of the worst generations) and 19 out of 40 incorrect generations. For DART, 24 generations are not correct. On WebNLG 2017, from the worst 40 generations, only two might benefit from improved fluency, while in many examples, the generated sentence was more fluent than the ground truth (14/40). Regarding correctness, only 2 out of 40 generations had a missing triple, while two generations incorrectly used a predicate. On WebNLG2020, only one instance exhibits room for improvement in fluency, but 24 instances either omit factual information or contain incorrect facts. Among the 20 outputs with the lowest BLEURT scores, 9 are rephrased texts with correctly explained facts. In contrast, among the 20 outputs with the lowest `FactSpotter`, only 4 instances fall into this category.

Based on our manual annotations, we observed that models are able to produce correct generations. However, when the generation is rephrased in respect to the ground truth sentence, metrics like BLEURT, which measure if two sentences are equivalent, struggle to assign high scores. We recall that BLEURT, a normalized metric, gives a score of 1 to equivalent sentences. On the dataset WebNLG 2017, our metric assigns a very high score to the models, while the highest average BLEURT score

is 73.44%. The BLEURT scores of the generations vary from 0.46 to 0.99; more than 50% of the test set generations score less than 0.80. We sampled 40 generations with BLEURT score $< 0.8$ and note that 35 generations are correct, which are rephrasing the ground truth, while 2 out of 35 that are better than the ground truth. Hence, we observe that BLEURT score cannot be used to determine if we have achieved a good performance on a dataset, it can only be used to compare different models. This issue has also been pointed out by the authors[4].

`FactSpotter` answers whether a fact is present in text; it does not have to address the much harder task of deciding if two sentences are equivalent. Besides being more reliable because it is solving a simpler task, it is also more interpretable as we can investigate the exact triples that are classified as negative, instead of doing a complete comparison between a subgraph and a sentences or between two sentences. This is especially useful for long input graphs and long generations.

## 8 Conclusion

In this work, we have presented a new metric for measuring factual faithfulness in G2T, `FactSpotter`. This metric can be trained in a self supervised fashion, using the same annotations as a G2T model. We have shown it achieves the highest correlation with humans on factual faithfulness and it can be used as a plug-in feature for G2T models. Finally, we have used our metric to analyze the difficulty of existing datasets. We have observed that models perform very well on these datasets, hence new datasets should be proposed. Such datasets could cover more difficult input graphs, for example triple from tables. In addition, through the initiative of governments to release tabular data related to public interest[5], tools trained to express in natural language the content of tables could be used as user friendly interfaces for citizens.

**Acknowledgment.** This work was performed using HPC resources from GENCI-IDRIS (Grant 2023-AD011014244). The authors were partially funded by the ANR-20-CHIA-0015 project.

---

[4] https://github.com/google-research/bleurt/issues/1
[5] For example, Eurostat, https://ec.europa.eu

## 9 Limitations

Our work has the following limitations:

- `FactSpotter` cannot be used to determine the precise nature of the error in the generated sentence. It was trained to predict whether a fact is presented in the text or not, not if we the sentence has a wrong predicate or a wrong subject or object. This problem can be solved by a second classification step for whether predicates or entities are incorrectly verbalized in the text, to make `FactSpotter` more interpretable.

- The input of `FactSpotter` is the concatenation of a fact $f$ represented in triple and a natural language text $T$, i.e., it has limited input format. With such input, the advantage is that it is easier to construct both positive and negative samples for its self-supervised training. However, it is also difficult to use it to check the factual faithfulness on other text generation tasks, since high quality structured knowledge graphs are hard to generate. However, we will investigate in the future the use of open information extraction models (Upadhyay et al., 2023) for extracting facts from sentences.

- Although the accuracy and F1 score of our classification model on the test splits of various datasets in Table 1 are high, there still exist some false positive and false negative samples. Hence, our `FactSpotter` generally reflects factual faithfulness, but it might still be biased on some hard samples, especially when predicates in knowledge graphs are distant to their natural language representations in the vector space of language models.

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

# A  Appendix

## A.1  Metrics for the evaluation of G2T

**BLEU (Papineni et al., 2002)**  This is one of the oldest sentence similarity metrics. BLEU is computed as a weighted geometric mean of $n$-gram precision scores, that is: it rewards inputs that share many common substrings (or $n$-grams, usually $n$ is 4). The score ranges between 0 and 1.

**METEOR (Banerjee and Lavie, 2005)**  This similarity metric is based on the harmonic mean of the 1-gram precision and recall. Its recall has a higher weight than precision. METEOR scores range between 0 and 1.

**BERTScore (Zhang et al., 2019)**  This leverages the pre-trained contextual embeddings of BERT, and matches words in candidate and reference sentences by cosine similarity. The score can be used to compute precision, recall, and F1. BERTScore is also a similarity measure, ranging from 0 to 1.

**BLEURT (Sellam et al., 2020)**  This is a fine-tuned Bert model on synthetically generated texts. It provides a similarity score learned from evaluation scores such as BLEU or ROUGE, and from human ratings; 1 represents a perfect match, while a value close to 0 means no similarity. We consider the authors' latest released model, BleuRT-20[6].

**BARTScore (Yuan et al., 2021)**  This metric is based on the BART seq2seq language model: the score represents the weighted log probability of one text $y$, given another text $x$. We compute two versions of the BARTScore, based on the indications of the paper: in the first version of the score, we compare the generated sentence with the gold standard (we refer to it as BARTScore, the default version provided by the authors), and a second version in which we compare the generated text with the input graph (referred in the paper as the faithfulness method, BARTScore-faithful in our experiments). The second score should encourage

---

[6] https://github.com/google-research/bleurt

the metric to consider the factual faithfulness of a generation. BARTScore is not normalized.

## A.2 Bootstrapping

Given two metrics $M_1, M_2$, we need to assess which one better correlates with human annotations. We have at our disposal a set of *G2T systems* $S_1, \ldots, S_S$, and a set $A$ of *G2T samples*. Given a set $D$ of *dimensions*, e.g., fluency, factfulness, etc., we have at our disposal the *human annotation* $H^d(S_i(a))$, $d \in D$ $1 \le i \le S, a \in A$: this is the score that human users assigned to the G2T generation of system $S_i$, on the sample $a$, along dimension $d$. Similarly, each automated metric $M$ computes $M^d(S_i(a))$, the $M$ score of $S_i$ on sample $a$ along dimension $d$. We note however that automated metric give generally just one score for all dimensions.

Based on the $H^d(S_i(a))$, $M^d(S_i(a))$ over all systems, samples, and dimensions, we assess how well metric $M$ correlates with human judgment $H$ in a statistically sound way. Since each dimension is of independent interest, we study the correlation of $M$ and $H$ *separately for each dimension*, and omit $d$ from the notations below, for simplicity.

We compute the three correlation coefficients ($r$, $\rho$ and $\tau$) between vectors of per-system scores $[M(S_1), \ldots M(S_S)]$ and $[H(S_1), \ldots H(S_S)]$. One aspect remains to settle: for a given $S_i$, how to aggregate information from all the annotated samples $A$? It is important to do this in a way that is robust to the divergence and noise sometimes found in human annotations.

As is proposed in (Dhingra et al., 2019), we apply *bootstrapping*, as follows. We generate a number (we used 1000, which is same as (Dhingra et al., 2019)) of "variants" $A_1, \ldots, A_{1000}$ of the annotated samples $A$, by sampling with replacement from $A$. Each $A_i$ has the same size as $A$, but it may contain some $A$ samples several times, and others not at all. We then compute correlation coefficients between the vectors $[M(S_1), \ldots M(S_S)]$ and $[H(S_1), \ldots H(S_S)]$ *over each $A_i$*, then represent each of the resulting 1000 correlation values by their *mean*, together a *confidence interval*. The relatively large number of "variants" acts as insulation against divergence and noise in the annotations. Then for two metrics $M_1, M_2$:

*i)* If the mean of $M_1$ is higher than $M_2$ and their confidence intervals do not overlap, it means $M_1$ *better correlates with human judgment, than $M_2$*.

*ii)* If the confidence intervals of $M_1, M_2$ overlap, we need to check whether a statistically significant conclusion can be drawn about the metrics. This is the case only if the confidence interval of the difference in correlation between the metrics is above 0 (Wilcox, 2016), in which the metric with higher mean correlates better with human judgment.

## A.3 Improved Samples of G2T Generation

Table 9 shows the improvement of factual faithfulness in G2T generation task after integrating FactSpotter into G2T inference. The samples are from the hardest zero-shot and compositional splits of GrailQA dataset. Each sample in the table includes a description of a subgraph, a ground truth text, a text generated by the best baseline model (JointGT), and a text generated by our FactJointGT.

Although the texts generated by the baseline model, JointGT, are generally fluent, we observe hallucinations in its generations: *i)* Some predicates are interpreted with different meanings in the generated texts; *ii)* Some entities are generated wrongly; *iii)* Some facts in subgraphs are lost in the generated texts. After integrating FactSpotter into the inference of G2T generation, without retraining the G2T model, our FactJointGT can generate texts that verbalize facts in subgraphs more correctly and completely. Hence the factual faithfulness of G2T task is improved by FactSpotter, but the other metrics do not necessary improve, since rephrased high quality texts might also be punished.

## A.4 Parameter tuning

### A.4.1 Percentages in negative samples for FactSpotter training

FactSpotter receives in the training set two types of negative samples. In the first type of negative sample (type I) we modify the fact from the graph, while the second type of negative sample (type II) we modify the text. To determine the best balance between negative samples of type I, respectively, type II, we compute the correlations between the score of FactSpotter and a subset of 50 annotations per system from the 2020 WebNLG challenge. As shown in Table 10, the best results for detecting factual accuracy and fluency are using either a ratio 0.9-0.1 or 0.8-0.2. In the remaining of the paper, we use FactSpotter with the parameters 0.9-0.1.

### A.4.2 $\lambda$ tuning for factual generations

We recall that in Equation 1, we have defined a weight $\lambda$ that quantifies the importance that we

| Subgraph | Ground Truth | JointGT Baseline | FactJointGT |
|---|---|---|---|
| (spacecraft, manufacturer, International Telephone & Telegraph), (spacecraft, manufacturer, Boeing Company), (spacecraft, count, ) | What is the total number of spacecrafts manufactured by International Telephone & Telegraph and Boeing Company? | International Telephone & Telegraph manufactures how many spacecrafts? | How many spacecrafts did International Telephone & Telegraph manufacture with Boeing Company? |
| (music recording, artist, Celine Dion), (music recording, length, float), (float, argmin, ) | What is the shortest Celine Dion song? | What is the musical recording that was recorded by Celine Dion? | Which musical recording was recorded by Celine Dion with the shortest length? |
| (John Elliott, games published, computer game expansion), (computer game expansion, count, ) | How many game expansions has John Elliott released? | How many video game expansions were published by John Ellison? | How many video game expansions were published by John Elliott? |
| (conference series, conferences of this type, seminar),(conference series, sponsoring organization, Australian Centre for Psychoanalysis), (conference series, question) | What conference series that has a sponsoring organization of Australian Centre for Psychoanalysis and is of type seminar? | Australian Centre for Psychoanalysis sponsors which conference series? | What conference series is sponsored by Australian Centre for Psychoanalysis and is a seminar? |
| (railway, terminuses, Strabane), (railway, question) | Strabane is the terminuses of which railway? | Which railway has a terminus of Stratane? | Strabane is the terminus of which railway? |

Table 9: Texts with improved factual faithfulness after integrating `FactSpotter` into G2T inference. Red stands for the contents that are presented in subgraphs, but are missed in the baseline generations. Blue stands for the exact contents correctly presented in ground-truth texts or FactJointGT generations.

| Type I / Type II | Data Coverage | | | Relevance| | | | Fluency | | |
|---|---|---|---|---|---|---|---|---|---|
| | $r$ | $\rho$ | $\tau$ | $r$ | $\rho$ | $\tau$ | $r$ | $\rho$ | $\tau$ |
| 0.9 / 0.1 | 0.85 | 0.84 | 0.68 | 0.93 | 0.88 | 0.75 | 0.91 | 0.79 | 0.63 |
| 0.8 / 0.2 | 0.83 | 0.85 | 0.63 | 0.92 | 0.93 | 0.79 | 0.89 | 0.79 | 0.61 |
| 0.7 / 0.3 | 0.77 | 0.84 | 0.66 | 0.85 | 0.89 | 0.73 | 0.66 | 0.63 | 0.50 |

Table 10: The influence of the ratio on `FactSpotter`.

| | IID | | Zero-shot | | Compositional | |
|---|---|---|---|---|---|---|
| $\lambda$ | BleuRT | FactS | BleuRT | FactS | BleuRT | FactS |
| 0.0 | 69.53 | 97.98 | 62.27 | 93.27 | **63.99** | 94.94 |
| 0.05 | **69.76** | 98.15 | 62.94 | 94.15 | 63.92 | 95.69 |
| 0.1 | **69.76** | **98.48** | 63.00 | 94.56 | 63.71 | 95.83 |
| 0.15 | 69.63 | 98.47 | **63.01** | 94.60 | 63.72 | **96.58** |
| 0.2 | 69.47 | 98.38 | 62.95 | **94.79** | 63.61 | 96.19 |

Table 11: BleuRT vs. `FactSpotter` when varying the $\lambda$ parameter (importance of the fact classifier for generating a sentence, see Equation 1).

assign to the probability that the next word is predicted such that we increase the probability of having a fact in the generated text, versus the default probability of choosing a word based on previously generated tokens. In Table 11, we vary the value

of $\lambda$ on the different splits of GrailQA on the T5 small model. The BleuRT score does not vary significantly, meaning that the fluency of the G2T model should not be affected. We fix $\lambda$ to 0.15 for the experimental evaluation in Section 6, as it gives the best performance on the zero shot split, which is the most challenging, and gives very good results also for the other splits. More precisely, we report the results of FactT5 and FactJointGT with $\lambda = 0.15$ on the datasets introduced in Section 6.3.

## A.5 Sentence level correlation

We explain two distinct definitions of sentence-level correlation provided by literature and report the correlations on the WebNLG human annotations.

The **first definition of sentence-level correlation** (Colombo et al., 2022) is outlined as follows:

- Construct a pair containing the automated metric scores $[M(\text{sys}_1), \ldots, M(\text{sys}_S)]$ and the corresponding system-level human scores $[H(\text{sys}_1), \ldots, H(\text{sys}_S)]$ for a given sentence.

- The Pearson correlation amidst these pairs is computed. If the correlation is significant (adopting $p < 0.05$ as the threshold), this correlation is preserved.

- Then, report the average over all significant correlations (average over at most the total number of annotated sentences).

When there is a large number of significant pairs with high correlation value, it can show if a metric can be used to compare different verbalisations of the same input triples. We report the human correlation results computed as this definition on the WebNLG 2017 and 2020 annotations in Tables 12 and 13 respectively. We note that only FactSpotter is significantly larger than the other metrics on data coverage and relevance for WebNLG 2020, and semantic adequacy for WebNLG 2017, which are metrics about factual faithfulness. Please note that here it is important to report on how many sentences the correlation was significant with $p < 0.05$. Because we compute if a metric has a higher score than a second metric using the bootstrapping technique presented in the Annexes, different samples might have different number of sentences with correlation having $p < 0.05$, hence we report the average number of sentences over all the samples. For WebNLG2017, out of 223 human annotated sentences, we have for semantics: BLEURT 171, FactSpotter 172, BERT 121 and BART 169. For fluency we have BLEURT 80 sentence pairs, FactSpotter 48, BERT 71 and BART 55. For textual structure BLEURT had 104 sentences pairs, FactSpotter 65, BERT 94 and BART 75. We note that for semantics the majority of pairs are significant (172 out of 223), meaning that FactSpotter can predict with high correlation improved verbalisations of the same input graph, hence it can be used to solve Q2. For WebNL2017, out of 178 human annotated sentences, we have: i) for correctness, BLEURT 121, FactSpotter 83, BERT 95 and BART 97 significant pairs on average; ii) for data coverage, BLEURT 102, FactSpotter 100, BERT 65 and BART 106; iii) for fluency, BLEURT 105, FactSpotter 49, BERT 90 and BART 72; iv) for relevance, BLEURT 99, FactSpotter 67, BERT 71 and BART 78; v) for text structure, BLEURT 92, FactSpotter 42, BERT 75 and BART 59; In this dataset, we obtain far less significant pairs, but we will show next that the annotators had more disagreement on WebNLG 2020.

| Metrics | Sem. Adeq. | T. Structure | Fluency |
|---|---|---|---|
| Human | 0.80 | 0.77 | 0.77 |
| BertF1 | 0.80 | 0.78 | 0.75 |
| BleuRT | 0.82 | 0.79 | 0.78 |
| BartS | 0.81 | 0.77 | 0.78 |
| FactS | **0.84** | 0.78 | 0.78 |

Table 12: The result of sentence-level correlation with the first definition on the WebNLG 2017 annotation.

| Metrics | Correct. | D. Cover. | Fluency | Relev. | T. Struct. |
|---|---|---|---|---|---|
| Human | 0.67 | 0.68 | 0.62 | 0.65 | 0.67 |
| BertF1 | 0.65 | 0.67 | 0.64 | 0.66 | 0.66 |
| BleuRT | **0.72** | 0.68 | **0.68** | 0.71 | **0.70** |
| BartS | 0.71 | 0.69 | 0.66 | 0.70 | 0.68 |
| FactS | 0.69 | **0.71** | 0.50 | **0.71** | 0.60 |

Table 13: The result of sentence-level correlation with the first definition on the WebNLG 2020 annotation.

| Metrics | Sem. Adeq. | T. Structure | Fluency |
|---|---|---|---|
| Human | 0.56 | 0.44 | 0.50 |
| BertF1 | 0.64 | 0.54 | 0.57 |
| BleuRT | **0.69** | **0.55** | **0.59** |
| BartS | 0.59 | 0.47 | 0.45 |
| FactS | 0.65 | 0.43 | 0.43 |

Table 14: The result of sentence-level correlation with the second definition on the WebNLG 2017 annotation.

| Metrics | Correct. | D. Cover. | Fluency | Relev. | T. Structure |
|---|---|---|---|---|---|
| Human | 0.38 | 0.37 | 0.29 | 0.35 | 0.30 |
| BertF1 | 0.42 | 0.38 | 0.42 | 0.37 | 0.40 |
| BleuRT | **0.45** | **0.43** | **0.47** | **0.38** | **0.43** |
| BartS | 0.39 | 0.38 | 0.32 | 0.35 | 0.31 |
| FactS | 0.38 | 0.40 | 0.29 | 0.37 | 0.27 |

Table 15: The result of sentence-level correlation with the second definition on the WebNLG 2020 annotation.

The **second definition of sentence-level correlation** (Banerjee and Lavie, 2005) is computed between the vector containing all the automatic scores for each sentence by a system $S$ given by a metric $M$, and the vector containing the human metrics for each sentence. The sentence-level correlation of a metric $M$ is computed as an average over all the correlation of each system $S_1, \ldots, S_S$. This measure determines if a metric can be used to rank verbalisations of different input graphs. The human correlation results computed as the second definition on the WebNLG 2017 and 2020 annotations are reported in Tables 14 and 15 respectively.

On WebNLG 2017 dataset, three metrics achieve similar scores for semantic adequacy, with no results significantly larger than the others (computed

| Metric | Krippendorff's Alpha |
|---------|---------------------|
| Fluency | 0.4201 |
| T. Struct. | 0.3231 |
| Sem. Adeq. | 0.5314 |

Table 16: Krippendorff's alpha for ordinal metrics on WebNLG 2017 human annotations

| Metric | Krippendorff's Alpha |
|---------|---------------------|
| Correct. | 0.2769 |
| D. Cover. | 0.2632 |
| Fluency | 0.2580 |
| Relev. | 0.1860 |
| T. Struct. | 0.2234 |

Table 17: Krippendorff's alpha for ordinal metrics on WebNLG 2020 human annotations

using confidence intervals by bootstrapping technique in Appendix A.2). Using (Evans, 1996) correlation guidelines, where a value between 0.6 to 0.79 is "strong correlation" and 0.8 to 1.0 is "very strong", we have "strong" correlation for the type of sentence correlation to the annotation. However, on WebNLG 2020 dataset, all metrics demonstrate a "moderate" level of correlation, given that the scores hover between 0.3 and 0.49.

We note that the two sentence correlation interpretations presented above are more sensitive to noise in the human annotation, as we consider each individual sentence score, not aggregates. This is evident also from the lower Human-Human correlation we observe on sentences, in comparison with the high correlation for the system level correlation. In Tables 16 and 17, we present the inter-annotator agreement for WebNLG 2017 and 2020 using Krippendorff's alpha (Krippendorff, 2018). It can be observed that both datasets exhibit generally low consistency among annotators, with the 2020 dataset showing less agreement than its 2017 version. However, on WebNLG 2017 we observe a higher agreement on semantic adequancy, for which we also observed a high correlation with our metric at sentence level.

We believe that these results show the need for better human annotation guidelines for this task. In particular, on WebNLG2020, the annotators were asked to give a score from 0 to 100 to a sentence for a given dimension such as correctness. Such a fine-grained decision is very difficult to take, hence the low agreement score. We note that mapping these scores to scores from 1 to 10 or 1 to 5 does

not improve agreement. In the 2017 challenge, the annotators gave scores from 1 to 3, but we believe that those scores were not sufficiently described such that the annotators could choose them accordingly. We recommend that future annotations use a Likert scale annotation, but with clearer guidelines and examples for each score.

### A.6 Qualitative Analysis for examples with different number of triples

We conducted a qualitative analysis on examples with different number of triples, to observe where the proposed metric shows improvements during decoding. We present the number of (graph, text) pairs as categorized by the number of triples within the input graph for the test set, and results for various datasets segmented by the number of triples are reported below. SimpleQuestion was not considered because it has only one triple per input graph.

For the WebNLG 2017 dataset, we can observe that for the small G2T model, T5S, the triples of size 4, 5 and 7 are more challenging, as is shown in Table 19. The integration of FactSpotter enhances the factuality of the results (as evident in F-T5S). We also observed a distinct improvement across various numbers of triples in the WebNLG 2020 dataset, especially when handling sentences comprising multiple triples.

As for the DART dataset in Table 23, inputs ranging from 1 to 5 triples witness improved results with FactSpotter's addition. Remarkably high FactSpotter scores are observed for inputs containing 6 or 7 triples. It's worth noting that of the inputs with 6 triples, 569 out of 598 are sourced from the E2E split, and for those with 7 triples, 320 out of 342 hail from the same E2E split. Hence, the E2E split of DART seems to be less challenging.

The results of GrailQA dataset are illustrated in Tables 25, 29, and 27. Regarding the GrailQA zero-shot split, single-triple verbalization consistently achieves superior scores with FactSpotter. Beginning with two triples, the incorporation of FactSpotter offers a discernible boost in model performance. In the compositional split, mirroring the trend observed in the zero-shot scenario, FactSpotter scores remain high for single-triple inputs. Furthermore, there is a marked improvement for inputs with two or more triples upon the addition of FactSpotter. In the IID split, while the baseline scores are impressively high, enhancements is noticeable for small model like T5S.

| Triple Num | 1 | 2 | 3 | 4 | 5 | 6 | 7 |
|---|---|---|---|---|---|---|---|
| Pair Num | 1540 | 821 | 419 | 241 | 185 | 0 | 6 |
| Percentage | 47.95 | 25.56 | 13.04 | 7.50 | 5.76 | 0 | 0.19 |

Table 18: The number and the percentage of (graph, sentence) pairs for the test set of WebNLG 2017 Const

| Triple Num | 1 | 2 | 3 | 4 | 5 | 7 |
|---|---|---|---|---|---|---|
| T5S | 99.87 | 99.45 | 99.13 | 97.62 | 98.09 | 97.53 |
| F-T5S | 99.75 | 99.65 | 99.22 | 98.74 | 99.20 | 99.93 |
| T5B | 99.65 | 99.69 | 99.35 | 99.15 | 99.52 | 99.93 |
| F-T5B | 99.81 | 99.70 | 99.69 | 99.85 | 99.53 | 99.93 |
| JGT-T5 | 99.69 | 99.73 | 98.52 | 98.63 | 99.30 | 99.93 |
| FGT-T5 | 99.40 | 99.89 | 99.50 | 99.89 | 99.60 | 99.93 |

Table 19: FactSpotter by the number of triples for the test set of WebNLG 2017 Const

| Triple Num | 1 | 2 | 3 | 4 | 5 | 6 | 7 |
|---|---|---|---|---|---|---|---|
| Pair Num | 369 | 698 | 1050 | 1220 | 1065 | 684 | 553 |
| Percentage | 9.09 | 17.19 | 25.86 | 30.04 | 26.21 | 16.84 | 13.62 |

Table 20: The number and the percentage of (graph, sentence) pairs for the test set of WebNLG 2020

| Triple Num | 1 | 2 | 3 | 4 | 5 | 6 | 7 |
|---|---|---|---|---|---|---|---|
| T5S | 93.71 | 93.91 | 92.24 | 90.71 | 89.95 | 88.68 | 86.12 |
| F-T5S | 95.05 | 95.13 | 95.58 | 94.07 | 92.21 | 91.01 | 90.54 |
| T5B | 95.21 | 96.33 | 94.51 | 93.00 | 92.29 | 92.30 | 93.36 |
| F-T5B | 96.27 | 96.74 | 95.69 | 94.41 | 94.72 | 94.89 | 94.09 |
| JGT-T5 | 94.74 | 94.71 | 92.71 | 92.16 | 88.16 | 89.10 | 88.46 |
| FGT-T5 | 94.26 | 95.85 | 93.85 | 93.93 | 90.90 | 91.43 | 92.48 |

Table 21: FactSpotter by the triple number for the test set of WebNLG 2020

| Triple Num | 1 | 2 | 3 | 4 | 5 | 6 | 7 |
|---|---|---|---|---|---|---|---|
| Pair Num | 848 | 797 | 821 | 869 | 822 | 598 | 342 |
| Percentage | 16.64 | 15.64 | 16.11 | 17.05 | 16.13 | 11.73 | 6.71 |

Table 22: The number and the percentage of (graph, sentence) pairs for DART test set

| Triple Num | 1 | 2 | 3 | 4 | 5 | 6 | 7 |
|---|---|---|---|---|---|---|---|
| T5S | 98.47 | 93.55 | 93.35 | 92.45 | 94.01 | 99.57 | 99.82 |
| F-T5S | 99.71 | 97.21 | 96.41 | 94.82 | 96.68 | 99.88 | 99.82 |
| T5B | 98.08 | 95.53 | 94.54 | 94.44 | 96.03 | 99.88 | 99.99 |
| F-T5B | 99.54 | 96.81 | 96.32 | 96.16 | 97.19 | 99.74 | 99.87 |
| JGT-T5 | 98.49 | 94.36 | 93.61 | 93.44 | 94.33 | 99.82 | 99.80 |
| FGT-T5 | 99.51 | 96.43 | 95.83 | 94.78 | 95.82 | 99.77 | 99.89 |

Table 23: FactSpotter by triple number for DART

## A.7 Further analysis on the difficulty of graph-to-text generation

We consider the intricacy in verbalizing a given triple is predominantly influenced by:

1. The distance between entity names in the knowledge graph and those in natural text.

2. The difference between KG predicates and their equivalent natural language phrases.

| Triple Num | 1 | 2 | 3 | 4 |
|---|---|---|---|---|
| Pair Num | 1231 | 522 | 55 | 20 |
| Percentage | 67.34 | 28.55 | 3.00 | 1.09 |

Table 24: The number and the percentage of (graph, sentence) pairs for GrailQA zero-shot split

| Triple Num | 1 | 2 | 3 | 4 |
|---|---|---|---|---|
| T5S | 96.41 | 89.63 | 92.71 | 93.62 |
| F-T5S | 97.08 | 91.30 | 94.52 | 99.39 |
| T5B | 96.49 | 93.03 | 92.84 | 94.98 |
| F-T5B | 97.48 | 93.31 | 94.28 | 99.68 |
| JGT-T5 | 96.96 | 90.77 | 93.50 | 91.28 |
| FGT-T5 | 98.02 | 91.63 | 95.95 | 93.81 |

Table 25: FactSpotter by the number of triples for for GrailQA Zero-shot split

| Triple Num | 1 | 2 | 3 | 4 |
|---|---|---|---|---|
| Pair Num | 388 | 292 | 39 | 38 |
| Percentage | 51.25 | 38.57 | 5.15 | 5.01 |

Table 26: The number and the percentage of (graph, sentence) pairs for GrailQA Compositional split

| Triple Num | 1 | 2 | 3 | 4 |
|---|---|---|---|---|
| T5S | 98.47 | 94.89 | 95.53 | 88.15 |
| F-T5S | 99.14 | 95.77 | 98.12 | 94.69 |
| T5B | 98.64 | 96.23 | 77.78 | 93.27 |
| F-T5B | 99.25 | 96.35 | 93.80 | 94.36 |
| JGT-T5 | 98.84 | 95.59 | 89.73 | 91.02 |
| FGT-T5 | 98.92 | 96.62 | 98.79 | 94.85 |

Table 27: FactSpotter by the number of triples for GrailQA Compositional split

| Triple Num | 1 | 2 | 3 | 4 |
|---|---|---|---|---|
| Pair Num | 257 | 465 | 63 | 12 |
| Percentage | 32.65 | 58.34 | 7.90 | 1.50 |

Table 28: The number and the percentage of (graph, sentence) pairs for GrailQA IID split

| Triple Num | 1 | 2 | 3 | 4 |
|---|---|---|---|---|
| T5S | 97.75 | 97.98 | 98.82 | 96.96 |
| F-T5S | 97.96 | 98.73 | 98.83 | 99.20 |
| T5B | 99.55 | 99.31 | 99.38 | 99.77 |
| F-T5B | 99.55 | 99.47 | 99.38 | 99.77 |
| JGT-T5 | 98.09 | 99.12 | 98.31 | 99.77 |
| FGT-T5 | 98.72 | 99.35 | 99.31 | 99.77 |

Table 29: FactSpotter by the number of triples for GrailQA IID split

**Entity Generation**   Regarding the difficulty of generating correct entities, we have the following statistics on the WebNLG 2017 dataset (v2.1):

- 87% of input graph entities appear identical in its corresponding ground-truth texts.

- 7% of differences arise from special characters, e.g., "Motherwell F.C." in the input graph becomes "Motherwell FC" in the ground-truth, both being correct.

- 2% are due to variations in date or number formats, as observed with "2006-12-31" in the KG interpreted to be "December 31, 2006" in the ground-truth text.

- 1% emerge from linguistic differences between the source language and English, as with "Atatürk" in KG being written as "Ataturk" in reference text.

- Other variances often regard alternative entity designations, such as "United States" in the KG being abbreviated as "US" in specific ground-truth texts.

For the GrailQA dataset, rooted in Freebase, 99% of input graphs retain consistent entity names in their ground-truth sentences. Accurate entity verbalization is relatively straightforward for this dataset, but we observe hallucinations in the texts generated by baseline models.

Considering that most differences between KG and natural language entity names result from formatting nuances, we can easily address its consistency challenge using constrained beam search to ensure all entity names appear in the generated text. For beam sizes over 10, the text generation becomes accurate without compromising other metrics. However, for beam sizes under 10, such constraints tend to impede the fluency.

**Predicate Generation**   In the WebNLG 2017 dataset, 49% of predicates appear the same as in the KB in their corresponding ground-truth sentences. Using SBert, we computed an average similarity of 84% between each predicate and its nearest n-grams in the ground-truth.

For the GrailQA dataset, only 27% of predicates align perfectly with their KG representations in the ground-truth. The average similarity stands at 64%, highlighting a larger difference between FreeBase predicates and their natural language phrases.

**Summary of Difficulty**   Datasets like WebNLG (from DBPedia) and GrailQA (from FreeBase) present challenges on different fronts. The difficulty from datasets based on FreeBase is the distance between knowledge graph and natural language is much higher. However, WebNLG and DART datasets have more complex input subgraphs, which has more number of triples, while GrailQA only has input subgraphs with up to 4 triples.

We consider that promoting accurate predicate generation is more challenging problem than promoting the generation of correct entities, because predicates are much more often rephrased in sentences, which is harder to evaluate. Inspired by the efficacy of Constrained Beam Search in ensuring accurate entity generation, we designed FactSpotter to enhance the accurate production of rephrased facts, especially rephrased predicates to be correctly generated.