# OpenReview forum: "FactSpotter: Evaluating the Factual Faithfulness of Graph-to-Text Generation"
_EMNLP/2023/Conference — EMNLP 2023 Findings_

### Official Review · Reviewer_wiwF · 2023-07-27

**Typos Grammar Style And Presentation Improvements:** NA
**Soundness:** 4

**Excitement:**

4: Strong: This paper deepens the understanding of some phenomenon or lowers the barriers to an existing research direction.

**Missing References:**

NA

**Paper Topic And Main Contributions:**

The authors investigate factual faithfulness within graph-to-text generation and propose a new metric, FactSpotter, which determines whether a given RDF triple is present in a generated text and can also be integrated into a generative model to improve factuality in outputs. They also consider the remaining difficulty of the task for state-of-the-art systems.

Their metric, FactSpotter, is basically trained to perform binary classification of whether a triple is mentioned properly within a text. The real-valued score is the result of the final softmax layer in the network, averaged over the set of facts for some graph-text pair. Ultimately, this appears to correlate well with human judgements of semantic adequacy, data coverage, etc. and better than most existing metrics.

They encourage more factual generations by modifying the prediction step such that new tokens are more likely to be added to the output if they increase the likelihood of better representing some fact. I.e. tokens will be reweighted according to their contribution to factuality wrt some fact until it is probable that the fact is already represented in preceding output text. On most datasets, integrating FactSpotter into modelling in this way allows for better factuality scores without much degradation (and sometimes further improvement) of other metrics.

**Questions For The Authors:**

Could this be easily extended to data-to-text tasks that use different data representations to RDF?

**Reasons To Accept:**

This is a well presented paper, with the authors very clearly explaining their motivations and the steps they take to develop and test their metric. The steps taken are intuitive and appear to yield strong results in evaluating factuality for this task. Their proposal for a method of integrating the metric into the model during decoding to promote generations that better contribute to factuality wrt the triple that needs to be lexicalized is creative and seems to offer good improvements to performance with little downside (e.g. no need to retrain the generative model).

**Reasons To Reject:**

The fact that this is built specifically around RDF triples as fact representations means that its applicability is likely limited to this specific task. It is also limited to an exhaustive set of predicates that happen to exist in the training data.

**Reproducibility:**

5: Could easily reproduce the results.

**Reviewer Confidence:**

3: Pretty sure, but there's a chance I missed something. Although I have a good feel for this area in general, I did not carefully check the paper's details, e.g., the math, experimental design, or novelty.

---

> ### Author Rebuttal · Authors · 2023-08-29
>
> We thank the reviewer for the feedback and suggestions! Below, we reiterate the reviewer's point and present our answer.
>
> *Could this be easily extended to data-to-text tasks that use different data representations to RDF?*
>
> **Answer:** Our work can be easily extended to data-to-text tasks. The DART dataset on which we run experiments contains a table-to-text split in which tables were represented as sets of triples. There are known methods for transforming a relational dataset into RDF, e.g., the R2RML language (in paper "R2RML: RDB to RDF Mapping Language"). We can train a FactSpotter with a self-supervised approach for any data-to-text task, and use it to encourage factual generations. Our FactSpotter also has capability under zero-shot settings, as long as the format is similar to training data. We will comment on this in the final version of the paper.

---

### Official Review · Reviewer_NiRu · 2023-08-02

**Soundness:** 3

**Excitement:**

2: Mediocre: This paper makes marginal contributions (vs non-contemporaneous work), so I would rather not see it in the conference.

**Missing References:**

- Modeling Global and Local Node Contexts for Text Generation from Knowledge Graphs (Ribeiro et al., TACL 2020)

**Paper Topic And Main Contributions:**

This paper presents a metric that aims to capture the faithfulness of a generated text compared to the fact present in a given triple. This metric is realized using a classifier based on a pretrained LM and trained with pairs or triples and texts with both positive and negative instances (that are created synthetically from the corpus perturbing the original triple or corresponding text).

The method is evaluated on different QA and Data-to-text datasets. It achieves strong correlation with human annotations for factors such as correctness, coverage and relevance. The paper also explores the use of the metric during decoding time to guide the generation to be more faithful to the input.

**Questions For The Authors:**

- I understand that the metric receives as input a triple and a text that contains the fact described in the triple but also describes other facts. Why don't the authors consider the whole graph as input instead of using a unique triple? Having a unique input for the graph would benefit the decoding?
- Do you have some thoughts on why the T5 base models have worse performance on FactS compared to small models (Table 4)? I would expect that larger models would generate more faithful texts to the input graph.

**Reasons To Accept:**

- This work focuses on measuring and improving factual consistency of graph-to-text models, which is an important factor when realizing structure data (e.g., triples into natural language).

**Reasons To Reject:**

- It would be good to see ablation studies and qualitative analysis for examples that the proposed metric improves during the decoding. For example, in examples that contains more facts (e.g., 5 triples) the model might have more difficulty generating factually generated text. Investigating such cases could bring insights and better illustrate the metric capabilities.
- Relying only on automatic metrics might restrict the results, especially regarding factual consistency which may not be well captured by metrics such as Bleu and METEOR. While the paper executes human inspections on factual consistency, it would be useful to illustrate generated (hallucinated) texts and factual scores exploring distinct dataset.

**Reproducibility:**

5: Could easily reproduce the results.

**Reviewer Confidence:**

4: Quite sure. I tried to check the important points carefully. It's unlikely, though conceivable, that I missed something that should affect my ratings.

**Typos Grammar Style And Presentation Improvements:**

- Factual faithfulness does not seem a little verbose to describe the factual quality of the generated text. Maybe factual consistency or simple faithfulness would be some suggestions.

---

> ### Author Rebuttal · Authors · 2023-08-29
>
> We thank the reviewer for feedback and suggestions! Below, we reiterate the reviewer's points and present our answers.
>
> ### Questions For The Authors:
>
> *1. I understand that the metric receives as input a triple and a text that contains the fact described in the triple but also describes other facts. Why don't the authors consider the whole graph as input instead of using a unique triple? Having a unique input for the graph would benefit the decoding?*
>
> **Answer:** We initially also considered promoting facts that can be inferred from the knowledge graph, by combining a reasoning model with our FactSpotter. However, we observed that inferred information is not always mentioned in the golden texts in datasets. For example, in WebNLG dataset, given input subgraph
>
>     Aaron_Bertram | associatedBand/associatedMusicalArtist | Kids_Imagine_Nation
>     Aaron_Bertram | associatedBand/associatedMusicalArtist | Suburban_Legends
>     Suburban_Legends | bandMember | Brian_Robertson_(trombonist)
>     Suburban_Legends | genre | Pop_music
>
> and ground truth text
>
>     Aaron Bertram plays for the Kids Imagine Nation band and the pop group Suburban Legends.
>     Brian Robertson is a member of the Suburban Legends and plays trombone.
>
> We can infer that Aaron Bertram and Brian Robertson are colleagues from the knowledge graph, but in the ground-truth sentences, this relation is not mentioned. Hence, we decided to use FactSpotter to promote only explicit facts from the input. Since we already observed that many single facts from input subgraphs are not in the texts generated by baseline models, we consider this contribution is valuable.
>
> In addition, the advantages of testing whether each triple is expressed well in a generated sentence are:
> (1) It makes the metric more **explainable**, since each atomic classifier can tell whether each triple is lost in a generated sentence or not; giving the whole graph as input to FactSpotter does not allow this.
> (2) It makes it easy to **embed FactSpotter into decoding** for more factual generation, as the reviewer mentioned. We can check whether each fact in each triple is already explained or not in the generation process, and then encourage the generation of each not included fact dynamically, according to Equation 1. For the facts that are already mentioned in the previously generated sequence, we no longer promote related information. Our goal is to _prevent generating repeatedly the same facts_, which is more useful than _always encouraging the generation of all facts in the graph_.
>
>
> *2. Do you have some thoughts on why the T5 base models have worse performance on FactS compared to small models (Table 4)? I would expect that larger models would generate more faithful texts to the input graph.*
>
> **Answer:** We believe that the larger models are **overfiting on SimpleQuestions**. The SimpleQuestions dataset is very large, but it has only 1-triple input graphs, many of which are about the same entities. The models might learn combinations of tokens during training, rather than interpreting knowledge graphs correctly. We have observed, for example, that "Nintendo" is often predicted as a game company by baseline models, even if the original input triple referred to another game company.
>
> ### Reasons to reject
>
> *1. It would be good to see ablation studies and qualitative analysis for examples that the proposed metric improves during the decoding. For example, in examples that contains more facts (e.g., 5 triples) the model might have more difficulty generating factually generated text. Investigating such cases could bring insights and better illustrate the metric capabilities.*
>
> **Answer:** We thank the reviewer for the useful suggestion. Indeed the cases with more input triples are more challenging: this is where the best improvements are obtained by pluging  our metric in a G2T. We report the results per number of triples for the datasets below (with just 1 triple per input graph, SimpleQuestion is not considered). We will add these results to the paper.
>
> **WebNLG17 Const Test**
> The number of  (graph, sentence) pairs, per number of triples in the input graph, in the test set:
>
> |Triple Num|1|2|3|4|5|6|7|
> |-|-|-|-|-|-|-|-|
> |Number of pairs|1540|821|419|241|185|0|6|
> |%age of pairs|47.95|25.56|13.04|7.50|5.76|0|0.19|
>
> It happens that no graph consists of exactly 6 triples.
>
> FactSpotter by size of the graph (number of triples):
>
> |Triple Num|1|2|3|4|5|7|
> |-|-|-|-|-|-|-|
> |T5S|99.87|99.45|99.13|97.62|98.09|97.53|
> |F-T5S|99.75|99.65|99.22|98.74|99.20|99.93|
> |T5B|99.65|99.69|99.35|99.15|99.52|99.93|
> |F-T5B|99.81|99.70|99.69|99.85|99.53|99.93|
> |JGT-T5|99.69|99.73|98.52|98.63|99.30|99.93|
> |FGT-T5|99.40|99.89|99.50|99.89|99.60|99.93|
>
> For WebNLG, we can observe that for the small model G2T, T5S, the triples of size 4, 5 and 7 are more challenging, and by pluging in FactSpotter we can improve their factfulness (F-T5S).
>
>
> **GrailQA**
>
>
> Zero-shot FactS Result
>
>
> | Triple Num      | 1     | 2     | 3    | 4    |
> | --------------- | ----- | ----- | ---- | ---- |
> | Number of pairs | 1231  | 522  | 55  | 20    |
> | %age of pairs   | 67.34 | 28.55 | 3.00 | 1.09 |
>
>
> FactSpotter by size of the graph (number of triples):
>
> | Triple Num | 1     | 2     | 3     | 4     |
> | ---------- | ----- | ----- | ----- | ----- |
> | T5S        | 96.41 | 89.63 | 92.71 | 93.62 |
> | F-T5S      | 97.08 | 91.30 | 94.52 | 99.39 |
> | T5B        | 96.49 | 93.03 | 92.84 | 94.98 |
> | F-T5B      | 97.48 | 93.31 | 94.28 | 99.68 |
> | JGT-T5     | 96.96 | 90.77 | 93.50 | 91.28 |
> | FGT-T5     | 98.02 | 91.63 | 95.95 | 93.81 |
>
> Overall, 1-triple verbalisations receive higher FactSpotter score; from 2 triples on, plugging FactSpotter improves the model performance.
>
>
> Compositional FactS Result
>
> | Triple Num      | 1     | 2     | 3    | 4    |
> | --------------- | ----- | ----- | ---- | ---- |
> | Number of pairs | 388  | 292  | 39  | 38    |
> | %age of pairs   | 51.25 | 38.57 | 5.15 | 5.01 |
>
> FactSpotter by size of the graph (number of triples):
> |Triple Num|1|2|3|4|
> | --------------- | ----- | ----- | ---- | ---- |
> |T5S|98.47|94.89|95.53|88.15|
> |F-T5S|99.14|95.77|98.12|94.69|
> |T5B|98.64|96.23|77.78|93.27|
> |F-T5B|99.25|96.35|93.80|94.36|
> |JGT-T5|98.84|95.59|89.73|91.02|
> |FGT-T5|98.92|96.62|98.79|94.85|
>
>
> Similarly to the zero shot case, we have high FactSpotter scores for 1 triples input, and we observe bigger improvements from 2 triples on when adding FactSpotter.
>
> IID FactS Result
>
>
> | Triple Num      | 1     | 2     | 3    | 4    |
> | --------------- | ----- | ----- | ---- | ---- |
> | Number of pairs | 257  | 465  | 63  | 12    |
> | %age of pairs   | 32.65 | 58.34 | 7.90 | 1.50 |
>
> FactSpotter by size of the graph (number of triples):
> |Triple Num|1|2|3|4|
> | --------------- | ----- | ----- | ---- | ---- |
> |T5S|97.75|97.98|98.82|96.96|
> |F-T5S|97.96|98.73|98.83|99.20|
> |T5B|99.55|99.31|99.38|99.77|
> |F-T5B|99.55|99.47|99.38|99.77|
> |JGT-T5|98.09|99.12|98.31|99.77|
> |FGT-T5|98.72|99.35|99.31|99.77|
>
> As for WebNLG, we observe on the IID split that only the small model benefits from the FactSpotter pluging (T5S compared with F-T5S), in particular on splits of size 2 and larger.
>
>
> **Dart**
>
> |Triple Num|1|2|3|4|5|6|7|
> |-|-|-|-|-|-|-|-|
> |Number of pairs|848|797|821|869|822|598|342|
> |%age of pairs|16.64|15.64|16.11|17.05|16.13|11.73|6.71|
>
> FactSpotter by size of the graph (number of triples):
>
> | Triple Num | 1     | 2     | 3     | 4     | 5     | 6     | 7     |
> | ---------- | ----- | ----- | ----- | ----- | ----- | ----- | ----- |
> | T5S| 98.47 | 93.55 | 93.35 | 92.45 | 94.01 | 99.57 | 99.82 |
> | F-T5S| 99.71 | 97.21 | 96.41 | 94.82 | 96.68 | 99.88 | 99.82 |
> | T5B| 98.08 | 95.53 | 94.54 | 94.44 | 96.03 | 99.88 | 99.99 |
> | F-T5B| 99.54 | 96.81 | 96.32 | 96.16 | 97.19 | 99.74 | 99.87 |
> | JGT-T5| 98.49 | 94.36 | 93.61 | 93.44 | 94.33 | 99.82 | 99.80 |
> | FGT-T5| 99.51 | 96.43 | 95.83 | 94.78 | 95.82 | 99.77 | 99.89 |
>
> For inputs of 1 to 5 triples, all models improve when adding FactSpotter.
> For inputs of 6 or 7 triples, we observe very high FactSpotter scores. For inputs of size 6, 569 out of 598 triples are from E2E split; for triple num 7, 320 out of 342 are from E2E split. E2E split only has triples about restaurant ratings, while the instances with less triples belong to several different datasets. We will provide in the final paper a more detailed analysis of DART, by dataset and number of triples in the input.
>
>
> *2.  Relying only on automatic metrics might restrict the results, especially regarding factual consistency which may not be well captured by metrics such as Bleu and METEOR. While the paper executes human inspections on factual consistency, it would be useful to illustrate generated (hallucinated) texts and factual scores exploring distinct dataset.*
>
>
> **Answer:** While BLEU and METEOR cannot measure factfulness, thus cannot reflect the effectiveness of our work, we report them for consistency with prior works ("JointGT: Graph-Text Joint Representation Learning for Text Generation from Knowledge Graphs", and "Investigating Pretrained Language Models for Graph-to-Text Generation"). We also report BLEURT, BART and Bert-F1 as SOTA metrics.
> We show **examples of hallucinations and how our method improves G2T models** in the Annexes, Table 8. The types of hallucinations that we have observed and that our work can address are the following:
>  - wrong entities in subject or object
>  - missing entities in subject or object
>  - wrong predicate
>  - missing predicate
>  - missing triple
>
>  This is possible through the construction of the negative set described in Section 4. We plan to devote more space to illustrating such hallucinations, in our final version.
>
> ### Missing references
> We thank the reviewer for pointing out the missing reference.
> In “Modelling Global and Local Node Contexts for Text Generation from Knowledge Graphs”, the authors designed a network which encodes both global and local information for text generation. JointGT, one of the baselines we used, can also represent both global and local information for data-to-text generation. The global information is from the original attention module of Transformers. Apart from that, the Structure-Aware Self-Attention Layer in JointGT can also capture local information.  Although various baselines designed neural networks to encode both global and local information, this does not guarantee that all facts in the input knowledge graph are correctly verbalised in the generated text. Hence, we think it’s useful to use FactSpotter to choose a better beam when decoding.
> We will add this discussion in our paper.
>
> ### Typos Grammar Style And Presentation Improvements
>
> We thank the reviewer's suggestions to the writing of our paper. We'll improve it according to the suggestions.

---

### Official Review · Reviewer_v9ZD · 2023-08-02

**Soundness:** 3

**Excitement:**

3: Ambivalent: It has merits (e.g., it reports state-of-the-art results, the idea is nice), but there are key weaknesses (e.g., it describes incremental work), and it can significantly benefit from another round of revision. However, I won't object to accepting it if my co-reviewers champion it.

**Missing References:**

- Entity-Based Semantic Adequacy for Data-to-Text Generation (Faille et al, 2021)
- Data-QuestEval: A Referenceless Metric for Data-to-Text Semantic Evaluation (Rebuffel et al, 2021)

**Paper Topic And Main Contributions:**

This work proposes a metric for establishing whether a knowledge base fact is correctly mentioned in a text. The method involves training a Electra (small) model with a binary classification objective on graphs and text from the training set of the WebNLG 2020 challenge. Evaluation was done by comparing the proposed metric's correlations (system-level) with human evaluation judgements from the 2017 and 2020 WebNLG graph-to-text (G2T) challenges. In addition, the authors propose applying their metric as a method to provide training signal when fine-tuning G2T models. They also claim that the results from their application of the metric to finetuning two model families (one based on pretrained language models with linearisation of the graphs, and another has a graph neural network component to model the  structure of the graph input) show that four existing G2T datasets, SimpleQuestions, GrailQA, WebNLG and DART are no longer challenging.

**Questions For The Authors:**

A: Line 273: How exactly is the FactSpotter score computed? Is it a sum of the binary 1, 0 prediction for each fact, or is it the sum of their probabablities (i.e. values after the softmax)? If it is the latter, won't your metric not be able to distinguish between a text where FactSpotter is moderately confident that all the facts are expressed in the text (i.e. 0.5 for each fact) versus another text where it is confident that half of the facts are not expressed in it?

B: Section 6 - Do you retrain specific FactSpotter models for your generation experiments in this section, i.e. using the training data for each of the datasets you evaluate against (SimpleQuestions, GrailQA, WebNLG an DART)? Or do you reuse the FactSpotter model trained on WebNLG 2020 data in Section 5 for all these other datasets? If the latter case, the WebNLG 2017 test sets were incorporated into the WebNLG 2020 training data (https://synalp.gitlabpages.inria.fr/webnlg-challenge/challenge_2020/#data), i.e. this means that despite your use of the constrained split of the 2017 WebNLG test set, the FactSpotter would have seen all of the fact triples in the constrained test set. If so, this could explain why you achieve the highest automatics scores for the WebNLG test set (i.e. the FactSpotter model is leaking knowledge of the test set into the generation). In any case, your conclusion (Line 531) --- that WebNLG (or any dataset for that matter) is no longer a challenging dataset --- will benefit from more supporting analysis/evidence than merely based on it having the highest automatic scores obtained.

C: Line 549: select the "best FGT-T5 model...", best with respect to what, or based on what measure?

**Reasons To Accept:**

Methods for automatically validating the factuality of text generated from graphs is important as G2T models become increasingly available and the barriers of access to them continue to decrease. It is important to have tools that can help detect whether these G2T generations (or any generated text in general) have poor factuality.

**Reasons To Reject:**

While RQ1, RQ2 and RQ3 of the paper can be complementary, the method proposed/conclusions made by the authors in the sections on RQ2 and RQ3 hinges on RQ1 (i.e. their proposed metric). Although the proposed metric outperforms in the human evaluations correlations reported for RQ1 (Table 2 and 3 in Section 5), the correlation was computed on a **system** level. To use the proposed method from RQ1 to address RQ2 (i.e. as a source of training signal to improve G2T generation) and RQ3 (for assessing whether existing G2T datasets are no longer challenging), it is important to know the **sentence** level correlation with human judgements, in order to be able to support the claims and conclusions made by the authors for those RQs (Section 6 and 7). The paper also makes a number of broad sweeping statements which arguably are not sufficiently supported by their experimental set-up (see Questions to Authors).

**Reproducibility:**

5: Could easily reproduce the results.

**Reviewer Confidence:**

4: Quite sure. I tried to check the important points carefully. It's unlikely, though conceivable, that I missed something that should affect my ratings.

**Typos Grammar Style And Presentation Improvements:**

- Line 153: it might be better to separately state that BLEURT is fine-tuned on human ratings that does consider semantic adequacy (or factual faithfulness), but not only that (i.e. fluency, grammaticality too).
- Section 6.4 can benefit from better organisation, with headers and clearer writing to highlight what aspects of RQ2 is addressed. Currently it reads as a mix of a discusson of RQ2 and RQ3 (see Line 531), as well as a discussion of your error analysis of the generations.
- Section 6.2 can benefit from clearer labeling each of the models and describing what they are and how they differ from each other. This way it will allow the reader to more easily understand and interpret the results reported in Table 4,5,6,7.

---

> ### Author Rebuttal · Authors · 2023-08-29
>
> ## Reviewer v9ZD
>
> We thank the reviewer for the feedback and suggestions! Below, we reiterate the reviewer's points and present our answers.
>
> ### Questions For The Authors:
>
> *A: Line 273: How exactly is the FactSpotter score computed? Is it a sum of the binary 1, 0 prediction for each fact, or is it the sum of their probabablities (i.e. values after the softmax)? If it is the latter, won't your metric not be able to distinguish between a text where FactSpotter is moderately confident that all the facts are expressed in the text (i.e. 0.5 for each fact) versus another text where it is confident that half of the facts are not expressed in it?*
>
>
> **Answer:** Indeed, using binary results allows distinguishing sentences where FactSpotter gives 0.5 probability for all facts to exist, from sentences that lack half of the facts. While we only report probability averages, we implemented binary and probability versions of FactSpotter. We reported averaged probabilities to capture different positive results qualities, e.g., a generation with 0.9 probability may be more factual than another with 0.6 probability. We can provide both results in the final version of the paper, and in our publicly available code, we will also provide both options. Note that users can see both the score per fact (what is the probability that a triple is present in the sentence) and the average over all facts, making our score interpretable. In addition, when we use FactSpotter to improve the prediction of G2T models, we check the score per fact until it becomes positive. Hence we can distinguish a missing fact, from a fact with a moderate probability.
>
>
> *B: Section 6 - Do you retrain specific FactSpotter models for your generation experiments in this section, i.e. using the training data for each of the datasets you evaluate against (SimpleQuestions, GrailQA, WebNLG an DART)? Or do you reuse the FactSpotter model trained on WebNLG 2020 data in Section 5 for all these other datasets? If the latter case, the WebNLG 2017 test sets were incorporated into the WebNLG 2020 training data, i.e. this means that despite your use of the constrained split of the 2017 WebNLG test set, the FactSpotter would have seen all of the fact triples in the constrained test set. If so, this could explain why you achieve the highest automatics scores for the WebNLG test set (i.e. the FactSpotter model is leaking knowledge of the test set into the generation). In any case, your conclusion (Line 531) --- that WebNLG (or any dataset for that matter) is no longer a challenging dataset --- will benefit from more supporting analysis/evidence than merely based on it having the highest automatic scores obtained.*
>
>
> **Answer:** We were also concerned about leaking knowledge from the test set during training. Therefore, **we retrain FactSpotter for each dataset, using only the corresponding training split**.
> We have reported the results of the training in Table 1. We trained FactSpotter on the 2.1 version of WebNLG, which has the same split as WebNLG2017.
> For the 2020 WebNLG human annotations, we kept the 2017 model, which has seen less data than WebNLG 2020. We have trained FactSpotter on WebNLG2020 (version 3 of the WebNLG data) and we show the changes that we get in respect to the paper:
> | Metric  | Correct. | D. Cover. | Relev. |
> | ------- | -------- | --------- | ------ |
> | FactSpotter WebNLG2020 |    0.94      |    0.93      |    0.95 |
> |  FactSpotter WebNLG2017 (paper)     |   0.94      |   0.91        |   0.96     |
>
> In Table 5 and the evaluation of Section 6.4 and 7, we analysed WebNLG v2.1, with FactSpotter only trained on WebNLG V2.1. A different analysis is required for the version 3 of the dataset, WebNLG2020. We will make this clear in the final version of the paper.
>
>
>
> We agree with your concern that we cannot conclude that WebNLG is no longer challenging based on it having the highest automatic score; line 531 was related to the evaluation we perform in Section 7. We will rewrite that paragraph to reflect only the information contained in Table 5, as follows: "_Table 5 has the highest FactSpotter score from all datasets, which means that we observe the most factual generations on WebNLG, according to FactSpotter. We will perform a human evaluation of factufulness in WebNLG in the next section._"
>
>
> Regarding the comment on the evaluation of challenging datasets, we add the following observations which we will also add to the paper.
>
> The difficulty for verbalizing a given triple strongly depends on: 1) the distance of the **entity names** in the KG to the entity names in the natural language, 2) the distance from the **predicates** in KG, to the phrases in natural language.
>
>
> Regarding the difficulty of **generating correct entity names**, we have the following statistics on the **WebNLG** dataset (v2.1):
> - 87% of input graphs' entities appear identical in its corresponding ground-truth texts.
> - 7% differences are from special letters, such as commas, dots, quotation marks, and parentheses. For instance, “Motherwell F.C.” in input graph is given as “Motherwell FC” in ground-truth text. They’re both correct spellings.
> - 2% are from different date or number format, such as “2006-12-31” in knowledge graph is interpreted as “December 31 2006” in text.
> - 1% are from the difference between its original language and English. E.g., “Atatürk” from knowledge graph is written as “Ataturk” in golden text. Both entity names should be correct.
> - Other differences are same entity with different quotations, such as “United States” in KG is written as “US” in some ground-truth texts.
>
> In **GrailQA**, a dataset based on Freebase, 99% input graphs have exactly same entity names in ground-truth sentences. Then verbalising the entities correctly for this dataset is a easy task.
>
> Since the distance between entity names in KG and natural language are mostly from formating, which all contain correct information, the problem of entity name consistency can be easily solved by using constrained beam search to force all entity names to appear in the generated text (When beam size is over 10, it can generate sentences with correct entities without suppressing other metrics. When beam size is less than 10, the fluency of generation decreases after adding constraints). We had investigates this line of research in our early experiments, however **constrained beam search does not easily generalize to enforcing predicates, because predicates are usually rephrased** in natural language.
>
> For the difficulty of **generating correct predicates**, in WebNLG dataset, 49% of predicates already appear the same as in the KB in their corresponding ground-truth sentences. We compute the average similarity between each predicate and the most similar n-grams in ground-truth sentence, according to SBert. The average similarity is 84%.
>
> In GrailQA dataset, based on Freebase, only 27% predicates appear exactly the same as in the KB in ground-truth sentences. The average similarity between each triple, and their most similar n-gram in the corresponding ground-truth text is 64%. So FreeBase predicates have a larger gap to their corresponding natural language phrases.
>
> We note that the differences between datasets based on different KBs might be due to the underlying KBs or to the way the G2T datasets were created and predicates were chosen.
>
> The WebNLG (from DBPedia) and GrailQA (from FreeBase) are difficult on different dimensions. The difficulty from datasets based on **FreeBase** is **the distance between knowledge graph and natural language is much higher**. However, WebNLG and DART datasets have **more complex input subgraphs**, which has more number of triples, while GrailQA only has input subgraphs with up to 4 triples.
>
> We consider promoting correct predicate generation is a more challenging problem than promoting the generation of correct entities, because predicates are much more often rephrased in sentences, which is harder to evaluate. Inspired by Constrained Beam Search which could force all entities to be correctly generated, we train FactSpotter to promote rephrased facts, especially rephrased predicates to be correctly generated.
>
>
> *C: Line 549: select the "best FGT-T5 model...", best with respect to what, or based on what measure?*
>
> **Answer:**“best FGT-T5 model” is with respect to the FactSpotter metric. We will clarify this.
>
>
> ### Reason to reject
> *While RQ1, RQ2 and RQ3 of the paper can be complementary, the method proposed/conclusions made by the authors in the sections on RQ2 and RQ3 hinges on RQ1 (i.e. their proposed metric). Although the proposed metric outperforms in the human evaluations correlations reported for RQ1 (Table 2 and 3 in Section 5), the correlation was computed on a **system level**. To use the proposed method from RQ1 to address RQ2 (i.e. as a source of training signal to improve G2T generation) and RQ3 (for assessing whether existing G2T datasets are no longer challenging), it is important to know the **sentence level correlation** with human judgements, in order to be able to support the claims and conclusions made by the authors for those RQs (Section 6 and 7). The paper also makes a number of broad sweeping statements which arguably are not sufficiently supported by their experimental set-up (see Questions to Authors).*
>
> **Answer:** We thank the reviewer for the opportunity to clarify. We base RQ2 and RQ3 not only on the human correlation results, but also on the high results in Table 1, in addition to the human evalution in Section 6.4 and 7. However, sentence-level correlation indeed strengthens our contribution and we can add it in the final version of the paper. The literature provides *two definitions of sentence level correlation*, both of which are interesting in our setting.
>
> The **first definition of sentence-level correlation** (from the paper "InfoLM: A New Metric to Evaluate Summarization & Data2Text Generation") is:
>  - We construct a pair containing the automatic metrics scores [M(System_1), . . . M (System_S)] and the corresponding system-level human scores [H(System_1), . . . H(System_S)] for a sentence, and we compute the Pearson correlation between the pairs. If the correlation is significant (we take p<0.05), we save this correlation.
>  - Then, we report the average over all significant correlations (average over at most the total number of annotated sentences).
>
> When we have a large number of significant pairs with high correlation value, this can show if a metric can be used to compare different verbalisations of the same input triples. This is important for Q2, where we use FactSpotter to improve the verbalisation of each individual sentence.
>
>
> We report below the human correlation results computed as above on the WebNLG2017 annotation. These results support the findings on the 2.1 version of WebNLG that we used in sections 6.3,6.4 and 7.
>
> | Metric | Sem. Adeq. | T.Structure | Fluency |
> | ------ | ---------- | ----------- | ------- |
> | BertF1 |     0.80       |     0.78        |    0.75     |
> | BleuRT  | 0.82           |       0.79      |   0.78     |
> | BartS       |  0.81          |       0.77      |   0.78      |
> | FactSpotter |  0.84*          |    0.78         | 0.78       |
>
> We note that only FactSpotter is significantly larger than the other metrics on semantic adequacy, the factufulness metric. Please note that here it is important to report on how many sentences the correlation was significant with p<0.05. Because we compute if a metric has a higher score than a second metric using the bootstrapping technique presented in the Annexes, different samples might have different number of sentences with correlation having p<0.05, hence we report the average number of sentences over all the samples.
> Out of 178 human annotated sentences, we have for semantics: BLEURT 171, FactSpotter 172, BERT 121 and BART 169. For fluency we have BLEURT 80 sentence pairs, FactSpotter 48, BERT 71 and BART 55.
> For textual Structure BLEURT had 104 sentences pairs, FactSpotter 65, BERT 94 and BART 75. We note that for semantics the majority of pairs are significant (172 out of 178), meaning that FactSpotter can predict with high correlation improved verbalisations of the same input graph, hence it can be used to solve Q2.
>
> We now move to the **second version of the sentence level-correlation**, (defined in paper METEOR: An Automatic Metric for MT Evaluation with Improved Correlation with Human Judgments). This correlation is computed between the vector containing all the automatic scores for each sentence by a system S given by a metric M, and the vector containing the human metrics for each sentence. The sentence-level correlation of a metric M is computed as an average over all the correlation of each system S_1, ... S_S. This measure determines if a metric can be used to rank verbalisations of different input graphs. We report the results in the following table:
>
> | Metric | Sem. Adeq. | T.Structure | Fluency |
> | ------ | ---------- | ----------- | ------- |
> | BertF1 |     0.64       |     0.54        |    0.57     |
> | BleuRT  | 0.69           |       0.55      |   0.59     |
> | BartS       |  0.59         |       0.47      |   0.45      |
> | FactSpotter |  0.65          |    0.43         | 0.43      |
>
> We observe that the three metrics achieve similar scores for semantic adequacy, with no results significantly larger than the others (computed using confidence intervals via the bootstrapping technique described in the Annexes). Using (Evans, 1996) correlation guidelines, where a value between 0.60 to 0.79 is strong correlation and 0.80 to 1.00 is very strong, we have strong correlation for the second type of sentence correlation.
>
> We note that the two sentence correlation interpretations presented above might be more sensitive to noise in the human annotation, as we consider each individual sentence score, not aggregates. The inter-annotator aggrement on the human annotations was not reported in the original report of the WebNLG2017 data (WebNLG Challenge: Human Evaluation Results).
>
> ### Missing references
>
> Thank you for pointing out these references!
>
> In **“Entity-Based Semantic Adequacy for Data-to-Text Generation”** the authors develop a metric for detecting if subgraph entities were generated in a verbalisation of the subgraph. Going beyond, our FactSpotter metric (a) also takes into consideration the predicate (not only the entities) and (b) it can be plugged in a G2T model. As highlighted above, enforcing entities in G2T is easier than enforcing predicates mentions. We can report also this metric on the top performing G2T models in the final version for completeness.
>
> In **“Data-QuestEval”**, the authors use a question generation (QG) and question answer (QA) framework, to evaluate the quality of generated text by testing whether each question from the QG module can be answered by the QA module.
> FactSpotter's goal is similar, but: (a) FactSpotter requires less computational resources as we do not need to use question generation and answering models, and the computational cost of a simple classifier for each fact is smaller than that of the sequence-to-sequence question answer model; (b) our model is self supervised, thus it does not requires additional data to the G2T model; ( c) our metric can be pluged into a G2T to improve its generation.
>
> We will add this discussion in our paper.
>
> ### Typos Grammar Style And Presentation Improvements
>
> We thank the reviewer's suggestions to the writing of our paper. We'll improve it according to the suggestions.

---

### Meta-Review · Area_Chair_hAgX · 2023-09-19

**Recommendation:** 3

**Metareview:**

This paper proposes a new metric, FactSpotter, to evaluate factual faithfulness by how the generated text covers facts in the knowledge base (KB). FactSpotter uses a classifier based on a pretrained LM and trained with pairs or triples and texts with both positive and negative examples. By using ELECTRA, FactSpotter outperformed conventional metrics in correlation to human evaluation results on the 2017 and 2020 WebNLG graph-to-text (G2T) challenges. Due to the necessity of the knowledge base, there is a consideration that FactSpotter may be restricted to specific tasks. However, the authors' response to Reviewer wiwF's question refers to the extension of FactSpotter by converting tables into knowledge bases. Since this approach partially solves the problem, we can judge that the paper should be accepted as a findings paper.

---

### Decision · Program_Chairs · 2023-10-07

**Decision:**

Accept-Findings

**Comment:**

This paper proposes a new metric, FactSpotter, to evaluate factual faithfulness by how the generated text covers facts in the knowledge base (KB). FactSpotter uses a classifier based on a pretrained LM and trained with pairs or triples and texts with both positive and negative examples. By using ELECTRA, FactSpotter outperformed conventional metrics in correlation to human evaluation results on the 2017 and 2020 WebNLG graph-to-text (G2T) challenges. Due to the necessity of the knowledge base, there is a consideration that FactSpotter may be restricted to specific tasks. However, the authors' response to Reviewer wiwF's question refers to the extension of FactSpotter by converting tables into knowledge bases. Since this approach partially solves the problem, we can judge that the paper should be accepted as a findings paper.